# Scandium Recovery Methods from Mining, Metallurgical Extractive Industries, and Industrial Wastes

**DOI:** 10.3390/ma15072376

**Published:** 2022-03-23

**Authors:** Ali Dawood Salman, Tatjána Juzsakova, Saja Mohsen, Thamer Adnan Abdullah, Phuoc-Cuong Le, Viktor Sebestyen, Brindusa Sluser, Igor Cretescu

**Affiliations:** 1Sustainability Solutions Research Lab, University of Pannonia, 8200 Veszprém, Hungary; ali.dawood@mk.uni-pannon.hu (A.D.S.); yuzhakova@almos.uni-pannon.hu (T.J.); thamer.abdullah@mk.uni-pannon.hu (T.A.A.); sebestyen.viktor88@gmail.com (V.S.); 2Department of Chemical and Petroleum Refining Engineering, College of Oil and Gas Engineering, Basra University, Basra 61007, Iraq; 3Nanotechnology and Advanced Material Research Center, University of Technology, Baghdad 35109, Iraq; 11659@uotechnology.edu.iq; 4Chemistry Branch, Applied Sciences Department, University of Technology, Baghdad 10011, Iraq; 5Department of Environmental Management, Faculty of Environment, The University of Danang-University of Science and Technology, 54 Nguyen Luong Bang, Danang 550000, Vietnam; lpcuong@dut.udn.vn; 6Department of Environmental Engineering and Management, Faculty of Chemical Engineering and Environmental Protection, Gheorghe Asachi Technical University of Iasi, 73 D. Mangeron Street, 700050 Iasi, Romania

**Keywords:** scandium recovery, hydrometallurgy, red mud, solvent extraction, organophosphorus extractants

## Abstract

The recovery of scandium (Sc) from wastes and various resources using solvent extraction (SX) was discussed in detail. Moreover, the metallurgical extractive procedures for Sc recovery were presented. Acidic and neutral organophosphorus (OPCs) extractants are the most extensively used in industrial activities, considering that they provide the highest extraction efficiency of any of the valuable components. Due to the chemical and physical similarities of the rare earth metals, the separation and purification processes of Sc are difficult tasks. Sc has also been extracted from acidic solutions using carboxylic acids, amines, and acidic β-diketone, among other solvents and chemicals. For improving the extraction efficiencies, the development of mixed extractants or synergistic systems for the SX of Sc has been carried out in recent years. Different operational parameters play an important role in the extraction process, such as the type of the aqueous phase and its acidity, the aqueous (A) to organic (O) and solid (S) to liquid (L) phase ratios, as well as the type of the diluents. Sc recovery is now implemented in industrial production using a combination of hydrometallurgical and pyrometallurgical techniques, such as ore pre-treatment, leaching, SX, precipitation, and calcination. The hydrometallurgical methods (acid leaching and SX) were effective for Sc recovery. Furthermore, the OPCs bis(2-ethylhexyl) phosphoric acid (D2EHPA/P204) and tributyl phosphate (TBP) showed interesting potential taking into consideration some co-extracted metals such as Fe(III) and Ti(IV).

## 1. Introduction

Scandium, along with yttrium and lanthanide, is sometimes categorized as a rare earth element [1]. Scandium is rarely found in concentrated amounts because it readily substitutes by major elements such as iron and aluminum [2]. Instead, it is found sparingly in trace amounts in rocks containing ferromagnetic minerals, with an abundance of 5–100 mg/kg [3]. Because of its poor distribution and the difficulty of extraction, Sc is a rare and expensive metal [4]. In aqueous solutions, scandium exclusively occurs in the III oxidation state (Sc^3+^) that tends to hydrolyze and form polymeric species [5].

The scandium hydrated ions are Pearson’s hard acids, which mean that they prefer to form complexes with hard ligands, including hydroxide, fluoride, sulfate, and phosphate ions, due to their high charge [6]. Scandium is primarily used to produce aluminum–scandium alloys, which typically contain 2% scandium by weight [7]. These alloys are significantly stronger than the conventional high-strength alloys, with great grain refinement, high strength, no hot cracking in welds, and exceptional corrosion resistance [8].

Furthermore, it is used in some modern applications such as high-intensity metal halide lamps, analytical standards, electronic components, oil well tracer, and fuel cells [9].

The involvement of scandium in the development of solid oxide fuel cells (SOFC) is also very appealing due to the strong oxygen-ion conductivity of Sc_2_O_3_-stabilized ZrO_2_ materials [10,11].

According to Jeong et al. [12], the Sc(III) doping into the layered perovskite material PrBaCo_2_O_5+_α (PBCO) has a significant impact on the electrochemical performance. They found that Sc(III) doping improved the Goldschmidt tolerance factor and specific free volume of the layered perovskites, resulting in a more suitable crystalline structure for oxygen ion transport in the lattice. As a result, at 500 °C, the Sc(III) doped PBCO has a maximum power density of 0.73 W cm^−2^, which is 1.3 times higher than PBCO.

Thortveitite, euxenite, and gadolinite are scandium minerals that contain significant amounts of scandium [2,13]. It also coexists in low amounts in aluminum, cobalt, iron, molybdenum, nickel, phosphate, tantalum, tin, titanium, tungsten, uranium, zinc, and zirconium ores [1,4,13].

Ores with a scandium content ranging from 0.002 to 0.005% can be considered resources and can be exploited [1]. The countries with the most scandium resources are the United States of America, Australia, China, Kazakhstan, Madagascar, Norway, Russia, and Ukraine [4,14]. Scandium is mostly found in uranium, tantalum, aluminum, and zirconium ores in America; nickel laterite ores in Australia; iron, tin, and tungsten ores in China; uranium ores in Kazakhstan; pegmatite rocks in Madagascar and Norway; and iron ores in Russia and Ukraine [1,15,16].

The commercial applications of scandium have been limited due to a lack of consistent and long-term production as well as its high price [17,18]. In the form of scandium oxide, the global scandium output is around 2 tons per year [15].

The development of scandium production was given special emphasis in the Soviet Union, with two government directives adopted to encourage it (1960 and 1985). In the mid-1980s, the Soviet Union produced approximately 10 t Sc every year, compared with barely 500 kg produced by the rest of the globe [19,20]. Scandium ores with high scandium content can be recovered using pyro-metallurgical processes. The energy consumption, on the other hand, is substantial. Scandium is frequently enriched in residues, slags, tailings, and waste liquors and is primarily produced as a by-product of ore processing [21,22]. The easiest way to recover scandium is by precipitation, i.e., insoluble scandium compounds from scandium containing solutions [23]. Even in this case, the co-precipitation of the other metals does not allow solutions with large quantities of metals to be recovered [24,25].

Hydrometallurgical processes currently used for scandium recovery mainly involve leaching, solvent extraction, and precipitation [26]. The complications of scandium recovery flow-charts depend on the various types and number of contaminants [27]. The separation and recovery of metals such scandium from other metallic impurities using an appropriate technology resulting in zero waste would solve a number of environmental problems [28]. Producing pure scandium metal from low-grade ores, secondary mud, or bauxite residue with overall scandium content around 100 mg/kg is a challenging task [1].

Hydrometallurgical units, like leaching and separation, that dissolve the contents of scandium into the aqueous phase and recover it by different separation methods appear to be promising [1,29,30]. During the production of other metals, such as Ti, W, U, Al, Ni, Tl, and Nb, and dusts in magneto-vana-ilmenite, chlorination, and tungsten residues refinery and red mud, scandium has been recovered from residues, tailings, and liquid waste [31,32,33].

Over the last few decades, researchers have put a lot of effort into purifying leaching effluents to separate scandium [5,6,34,35,36,37,38,39,40,41]. The purification took place by solvent extraction (SX), ion exchange (IX), liquid membranes, adsorption, precipitation, and co-precipitation [42,43]. Because of its high extraction capacity, high selectivity, effective separation, high enrichment, and ability to operate on a large scale, solvent extraction is one of the most effective methods for the separation and purification of target metal ions from various aqueous solutions [44,45,46].

Many commercially available organophosphorus extractants were used to separate scandium from the aqueous phase [4,9,47]. Due to the higher extraction efficiency of the combined reagents, the SX procedures have proven to be more effective, allowing the high enrichment of trace metal ions such as those of rare earth elements (REE) [48,49].

Furthermore, changing the equilibrium pH in the SX system may result in a high selectivity towards the targeted rare metal ions, achieving the removal of impurities [50,51,52]. Low scandium concentrations are frequently recovered as an unwanted by-product during the treatment of tailings and residues from a wide range of sources, such as the uranium leaching process, titanium pigment manufacturing, liquid wastes, and chlorinating magneto-vana-ilmenite dust [1,53]. Solvent extraction and associated technologies such as ion exchange (IX) and liquid membrane (LM) extraction are extensively used for the separation and purification of scandium from diverse aqueous solutions [54,55].

For extracting trace amounts of scandium from solutions containing substantial levels of other elements, SX is currently the most extensively used technique [54,55]. Furthermore, it is the most extensively used method for scandium separation and purification because of its great extraction capacity and operation simplicity at large scales.

In this paper, the separation and purification methods of scandium using acidic, basic, neutral, and chelating extractants as well as synergetic solvent extraction procedures were discussed.

Furthermore, the metallurgical procedures for scandium recovery from a variety of sources and particularly from bauxite wastes (red mud) were among the primary objectives of the publication.

## 2. Separation and Purification of Rare Earth Elements Using SX

The process of recovering rare earth elements typically consists of three important steps that begin with the solid waste or ores: (1) acid leaching (convert the solid to the ionic form), (2) solvent extraction step to extract and enrichment the concentration of target metal ions, and (3) purification of the produced solution using various metallurgical processes such as precipitation and then calcination. Figure 1 presents a schematic flow diagram of the major stages in the REE processing [56,57,58,59].

### 2.1. Acidic Extractants

Organophosphorus, carboxylic, and sulphonic acids having the functional groups –POOH, –COOH, and –SO_3_H, respectively, are acidic extractants [60]. In general, when the organic phase contains a large concentration of metal ions (M^n+^), the extraction of a metal cation (M^n+^) by an acidic extractant (HA) occurs via a cation exchange process, as illustrated in Equation (1) [61].
(1)$$M^{n+}+n\;\bar{\mathrm{HA}}↔\bar{{\mathrm{MA}}_{n}}+n\;H^{+}$$

The mechanism, though, is difficult because of the creation of a dimer in the organic phase during the interaction. When there is a low metal loading in the organic phase, the extraction can be represented by Equation (2) [1,62].
(2)$$M^{n+}+n\;({\bar{\mathrm{HA})}}_{2}↔\bar{{\mathrm{MA}}_{n}\;.\;{\left(\mathrm{HA}\right)}_{n}}+n\;H^{+}$$

#### 2.1.1. Acidic Organophosphorus Extractants

As illustrated in Figure 2 phosphoric, phosphonic, and phosphine acids are the most used acidic organophosphorus extractants [1,63,64]. Di-(2-ethylhexyl) phosphoric acid is the most used extractant for concentrating scandium and separating it from other metals. Although it is commercially available under different names, including D2EHPA, HDEHP, P204, and DP-8R, we referred to it as P204 in this work [42,53,57].

In a previous work [51], the behavior of the extraction of certain transition metals in HCl, HClO_4_, and HNO_3_ media was investigated over an acidity range of 1–11 mol/L using 0.75 mol/L P204. Over 99% of the scandium was extracted without any interference of Mn^2+^, Tc^2+^, and Re^2+^ under all working conditions. On the other hand, Ti(IV), Zr(IV), and Hf(V) were extracted quantitatively and could interfere with the extraction of scandium. Following the extraction order Sc^3+^ > Fe^3+^ > Lu^3+^ > Lu^3+^ > Yb^3+^ > Er^3+^ > Y^3+^ > Ho^3+^, P204 can extract and separate among large amounts of iron and other elements in an acidic chloride medium [53].

At a pH of 0.35, almost 98% of the Sc(III) was extracted, whereas iron(III), yttrium(III), and lanthanides(III) were left in the solution. A study involving P204 as an extractant and tributyl phosphate (TBP) as a modifier to recover scandium from HCl-leaching solutions of magnesium, aluminum, and iron scraps showed that after a single stage of extraction and stripping, the scandium recovery was nearly 100%, and its separation from magnesium was nearly quantitative [16].

The significant co-extraction of impurities as well as the problems of stripping and emulsion generation limited the industrial of this method. 2-ethylhexyl phosphonic acid mono-2-ethylhexyl ester, characterized as HEHEHP in this paper, is another commonly used extractant available under trade names such as P507, PC-88A, Ionquest 801, and SME 418.

The extractability of HEHEHP was similar or somewhat lower than P204, while the required acidity of the strip solution was lower [65,66]. Sc(III) was quantitatively extracted by HEHEHP in HCl and HNO_3_ solutions of an acidity range of 1–5 mol/L with an extraction order of Sc(III) > Th > Ce(IV) > Lu(III). From H_2_SO_4_ solution of 1–5 mol/L, over 98% of the Sc(III) was recovered with an extraction order of Sc(III) > Ce(IV) > Th > La(III) and only about 20% of the Fe(III). The stripping efficiency of scandium from the HEHEHP extractant by mineral acids is more efficient than that of the P204. Compared with P204, the H_2_SO_4_ solution of 5 mol/L stripped about 20% of the Sc(III) from HEHEHP, whereas P204 stripped only 5% [35]. For Sc(III) extraction from acidic solutions by organophosphorus compound (OPCs), the iron is the most common interfering element because of its Lewis acid hardness and the amount of iron in the solution.

Zr, Ti, Th, Y, and Ln elements that are typically linked with scandium also impede the extraction of scandium to some extent. Previous research [16,42,55,56,57] focused on the extraction of Sc(III) using C272, C302, and C301, as well as the other compounds. In their findings, they observed that the scandium extraction efficiency using these three extractants is high at low acidities, with the metal selectivity in the following order: Zr(IV) > Sc(III) > Fe(III) > Lu(III). The scandium recovery from C272, C302, and C01 was, likewise, considerably easier than this from P204 and HEHEHP [42,55].

After one SX with the C272, C302, and C301 extractants, the scandium recovery efficiency was about 82%, 78%, and 75%, respectively, with concentrations of sulfuric acid of 1.5, 3.5 and 5.8 mol/L [16]. Using P204, Ionquest 801, and C272 to study the recovery of scandium from a laterite waste solutions, the results related to their recovery were similar to those presented in other papers [55].

For scandium extraction from C272, a recovery rate between 90% and 99% was achieved when 300–400 g/L sulfuric acid was used, while the corresponding from P204 and Ionquest 801 was more difficult. The following extraction order also observed in the case of use of C302 to separate scandium in chloride media: Sc(III) > Y(III) > La(III) > Gd (III). Moreover, nitric acid (of 5 mol/L and 4 mol/L) can completely recover Sc(III) and Y(III) from the C302 organic phase, respectively [16]. Direct scandium recovery with the formation of insoluble precipitates by alkaline or fluoride salts is an alternative strategy for processing. As an illustration, 98% of the Sc(III) was stripped directly from the loaded P204 with 3 mol/L sodium hydroxide. Conversely, the recovery with low-to-medium concentrations of sodium hydroxide (0.25–2.5 mol/L) produced phase separation difficulties [17].

The use of stronger alkaline solutions, such as 5 mol/L sodium hydroxide, can alleviate the phase separation problem to some extent because the solubility of the organic solvent in the loaded strip liquid is reduced [13]. Using NaF solutions of 2 mol/L, scandium can be completely removed from the organic phase of P204 and HEHEHP [38]. The scandium stripping technique by ScF_3_ precipitation is effective, but the fluoride, because of its toxicity, could have a negative impact on the environment. The mechanism of acidic phosphorus-based extraction changes with the acidity of the aqueous solution [49,53,67].

At low acidities, the acidic extractant molecules lose protons to form Sc-organic complexes by a cation exchange mechanism, but at high acidities, Sc is extracted via a solvation mechanism. As an illustration, the distribution ratio of scandium during the SX with C272 decreased for a sulfuric acid concentration of 1.5 mol/L due to the cation exchange of the recovered species Sc(A)_3_·2HE [53]. The extraction efficiency of Sc improved with increasing concentration due to the formation of the extracted species HSc(SO_4_)_2_·3HE. Furthermore, the organic molecules were expected to be neutral ligands, meaning that the extraction mechanism was solvation.

**Figure 2 materials-15-02376-f002:**
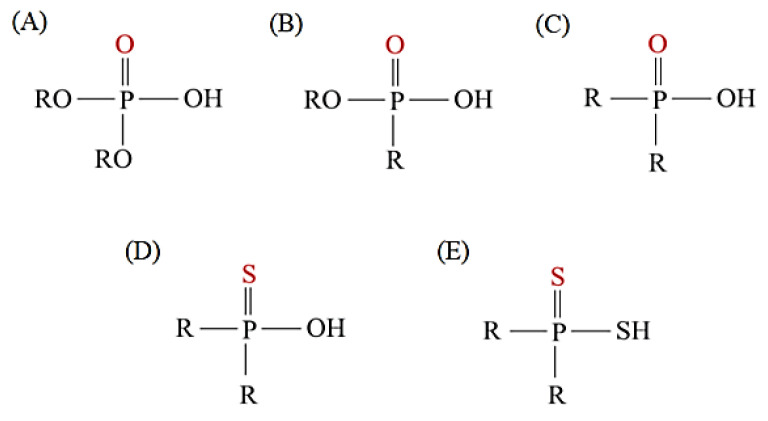
Chemical structures of the acidic phosphorus extractants: (**A**)—phosphoric acid, (**B**)—phosphonic acid, (**C**)—phosphinic acid, (**D**)—monothiophosphinic acid, and (**E**)—dithiophosphinic acid, adapted from [9,68].

#### 2.1.2. Carboxylic Acids

Scandium is commonly extracted from low-acid feed solutions using carboxylic acids, such as versatic, phenoxy acetic, and naphthenic acids (Figure 3). Alkyl monocarboxylic acids, such as naphthenic and versatic acids, are commercially available (i.e., V10). In REE separation, naphthenic acids or related derivatives of carboxylic acid were widely used. Naphthenic acids preferentially extracted uranium and thorium over trivalent rare earth elements [59]. It was shown that the separation factors of yttrium from lanthanides aqueous solutions improved when chelating agents e.g., ethylenediaminetetraacetic acid (EDTA) were used. This metal extraction was achieved by using undue concentrations of naphthenic acids (1 mol/L) in kerosene with an A/O ratio of 4 [60].

In the literature [70], 5–10% carboxylic acid was used to extract Sc from AlCl_3_ solutions containing 20–30 g/L iron. Sc was recovered using a combination of hydrochloric acid of 3 mol/L and sulfuric acid of 1.5 mol/L. The selectivity of scandium over all REEs was quite high (β of Sc/REEs > 104) when kerosene containing a naphthenic acid and iso-octanol were used as the extractants and HCl as the stripping agent. In this case, the extraction sequence was Sc > Sm > Eu >Y > Nd > La > Yb > Gd [71].

However, the appropriate pH for the Fe(III) removal by naphthenic acid is 3 in comparison with approximately 2.8 for scandium, which indicates an interference of iron and scandium in the extraction process. An emulsion was formed when iron was completely extracted at a pH of 3.8, leading to a slow phase separation [72].

Consequently, before extracting scandium by naphthene acid, iron should be removed from the feed solution. Because aluminum was totally extracted at a pH of 4, it should also be eliminated before the extraction of scandium. Sc(III) is extracted more efficiently by alkyl-phenoxy acetic acid (APAA) at lower pH (ca. 2.4), indicating that, in this way, it can also be efficiently separated from Fe(III) and Al(III) rather than by naphthenic acid [46,73].

With sec-nonylphenoxy acetic acid (CA-100) in n-heptane, the extraction behavior of trivalent REEs and bivalent metal ions (Cu, Zn, Ni, Mn, Cd, and Co) from HCl solutions was investigated [46]. There were high separation factors (β) of Sc over Y, La, and all bivalent metals. For example, the separation factor for Sc/Y was 407, Sc/La 83, Sc/Gd 89, and Sc/Lu 269, indicating that CA-100 is an efficient extractant for scandium [74].

The separation of scandium traces was enhanced for REE solutions of pH between 2 and 4 by using an extractant containing 0.2–1.0 mol/L of APAA and 5–30% long carbon chain alcohols [2].

In order to achieve a Sc-enriched strip solution, the loaded phase was stripped with HCl acid (0.5–3 mol/L). Using a counter current extraction, scandium was successfully re-extracted from the solution in the pH range of 1 to 4. Scandium of high purity (99.9%) and high scandium recovery (90%) were obtained using the two SX techniques described above. Cation exchange mechanisms such as ScA_3_ complex formation and HA molecule entry into the complex can both be used to extract scandium from low-acid solutions by carboxylic acids (HA) and produce adducts such as ScA_3_·(HA) as part of the extraction process [63]. Because of their steric bulk, carboxylic acid molecules had an impact on how well they extracted the lanthanide series elements from solution [65].

The previously mentioned carboxylic extractant V10 was found to be effective in extracting monomeric complexes of the LaA_3_·(HA)_3_ type, whereas with 3-(cyclohexylazaniumyl) propanoate acid and dimeric complexes, such as (LaA_3_ (HA)_3_)_2_, (GdA_3_ (HA)_3_)_2_ and (LuA_3_ (HA)_3_)_2_ were formed. Table 1 summarizes previous SX investigations on scandium recovery using a variety of extractants.

### 2.2. Alkaline Extractants

Primary amines (RNH_2_), secondary amines (R_2_NH), tertiary amines (R_3_N), and quaternary ammonium salts (R_3_N^+^CH_3_X^−^), where X^−^ is usually a halide ion, are all common alkaline extractants. Metal extraction by amines is thought to be primarily dependent on the ability of the metal ions to form anionic species in the aqueous phase, which are extracted by amines via anion exchange processes. Scandium from uranium mill waste solutions was successfully extracted using a primary aliphatic amine (Primene JMT), an amine extractant [18,50]. Primene JMT was used at 2.5% in kerosene in a small-scale pilot plant, and the A/O ratio obtained was 50:1. Although a minor amount of Fe(III) was co-extracted, scandium can successfully be separated from iron and easily be recovered with a 2 mol/L NaCl solution acidified to pH 1 at an A/O ratio of 1:10. In addition, alkaline stripping was efficient, but a Sc(OH)_3_ precipitate formed during the back extraction resulted in phase disengagement problems. The extraction of scandium from sulfuric acid solutions using a primary amine extractant N1923 (RNH_2_) was also investigated [84,85].

Scandium has a high distribution coefficient (k_D_) for a H_2_SO_4_ solution between 2 and 4 mol/L. The proposed reaction is shown in Equation (3).
(3)$$\mathrm{Sc}{\left({\mathrm{SO}}_{4}\right)}_{3}^{3-}+1.25\;(\bar{({\left({\mathrm{RNH}}_{3}\right)}_{2}{\left({\mathrm{SO}}_{4}\right)}_{2}})↔\bar{{\left({\mathrm{RNH}}_{3}\right)}_{3}\mathrm{Sc}{\left({\mathrm{SO}}_{4}\right)}_{3\;}\cdot {\left({\mathrm{RNH}}_{3}\right)}_{2}{\mathrm{SO}}_{4}}+1.5\;{{\mathrm{SO}}_{4})}_{4}^{2-}$$

The extraction of Sc from acidic sulfate solutions by bis(3,5,5-trimethylhexyl) (NH₄)₂SO₄ was found to have stoichiometry of (R_2_NH_2_)_4_ScOH(SO_4_)_3_, where R is 3,5,5-trimethylhexyl [28].

A previous paper [16] stated that the primary amine N1923 extracted Sc(III) as Sc(OH)^2+^ from thiocyanate and nitrate-thiocyanate solutions. In another investigation, scandium was quantitatively extracted from 0.1 mol/L of malonic acid (H_2_A) by 4% Amberlite LA-1 (secondary amine R_2_NH) at a pH of 2.4 to 5.4. Moreover, it was selectively recovered from the organic phase using 0.5 mol/L HCl, leaving it with the remaining impurity metals such as Cu(II), Fe(III), U(VI), and Ti(IV) [76].

Tri-benzyl amine, primene JMT, tributylamine, and tri-iso-octylamine were also investigated as extractants and found to be unsuitable for a variety of purposes. Among xylene, toluene, benzene, chloroform, carbon tetrachloride, hexane, cyclohexane, and kerosene tested as solvents, xylene was found to be the most efficient.

The following reaction was used to extract the soluble Sc(III)-malonate anion (A^−^) complex from the scandium solution (Equation (4)):(4)$$2\mathrm{Sc}{\left(A\right)}_{3}^{3-}+\bar{3({\left(R_{2}{\mathrm{NH}}_{2}\right)}_{2}\;A)}↔\bar{2{\left(R_{2}{\mathrm{NH}}_{2}\right)}_{3}\mathrm{Sc}{\left(A\right)}_{3}}+3A^{2-}$$

A previously published paper [70] investigated the separation of Sc and Y from Ln using different quaternary ammonium salts Aliquat 336, MAN, MDA, and TBHAN. The results showed that Aliquat 336 had the maximum extraction efficiency under the presented experimental conditions. However, the distribution ratios of lanthanides dropped according to their increasing atomic order. Scandium was found to be in the extraction sequence between samarium and gadolinium [78].

### 2.3. Solvating Extractants

Organic compounds containing –C=O and –P=O groups, such as ketones, ethers, and phosphates, are commonly used as solvating extractants. The extraction of elements is based on the solvation of neutral inorganic compounds or complexes by extractants containing electron donors [60]. By adjusting the pH, it was possible to achieve selective extraction of Sc(III) by mesityl oxide (4-methyl-3-pentene-2-one, MeO) from a sodium salicylate solution including Fe(III), Ti(IV), V(V), Bi(III), Cr(VI), Zr(IV), Mo(VI), La(III), and Th(IV) [86]. Sc(C_6_H_4_(OH)COO)_3_·3MeO is the most probable extracted species, and the extraction mechanism usually requires solvation of scandium salicylate. There are four types of neutral organophosphorus compounds that are often employed as solvating extractants: trialkyl phosphate, dialkyl-alkylphosphonate, dialkyl-alkylphosphinate, and trialkylphosphine oxide (Figure 4).

The neutral OPCs are presented in the following decreasing order of the electron density of the phosphoxyl group: phosphates, phosphonates, and phosphinates, which are followed by phosphine oxides. As a result, the strength of their association with metals decreases in the same manner as the preceding order [77].

TBP is commonly utilized to extract, separate, and concentrate REEs, including scandium. Based on the literature, the extraction of Sc and Th can be achieved in the concentration range of 7–8 mol/L HCl by using a concentrated TBP solution [87].

Nevertheless, due to the substantial co-extraction of Zr, the separation of Sc and Zr using TBP was not feasible. Furthermore, the high acidities required for high scandium extraction lead to various drawbacks, which can be detrimental. Because only solutions with HCl concentrations greater than 8 mol/L allow the complete extraction of Sc(III) and, simultaneously, the same conditions enable the total extraction of iron, the separation of Sc from Fe is challenging [81].

P350 was found to have a greater extraction efficacy than TBP for the extraction of scandium from hydrochloric acid solutions as well as a higher selectivity over a wide range of interfering elements, like iron. Sc(III) was recovered as ScCl_3_·3(P350) in n-heptane with the −P=O group of the phosphate incorporated, as shown in Equation (5):(5)$${\mathrm{Sc}}^{3+}+3\bar{P350}+3{\mathrm{Cl}}^{-}↔\bar{{\;\mathrm{ScCl}}_{3}\cdot 3\;P350}$$

Zhang and collaborators [78] described a technique to produce ultrapure Sc_2_O_3_. The procedure is divided into two SX cycles: the first cycle includes removal of Zr from the leachate (HClO_4_ solution of 6 mol/L) with concentrated TBP at an O/A phase ratio of 1.

In the second cycle, an organic phase containing 40 vol% of P350 in kerosene was used for the extraction of Sc and its separation from interfering metals such as Ca, Al, Mn, Ti, Y, and lanthanides (Ln) in three stages with an O/A phase ratio of 1.9:1. In these three steps, 1 mol/L HCl was used to entirely recover the Sc from the loaded organic phase.

Following precipitation with H_2_C_2_O_4_ and thermal treatment at 800 °C, approximately 94% of the scandium was recovered as Sc_2_O_3_. The extraction of scandium from aqueous solutions (leachate) containing high concentrations of HCl and H_2_SO_4_ acids was investigated using C923 and C925. C923 and showed high scandium extraction in the concentration range between 2.0–7.0 mol/L H_2_SO_4_ with an excellent separation factor for Fe [70].

Scandium and iron extraction was significant with C923 in an acidic medium of concentration lower than 1 mol/L H_2_SO_4_, resulting in unsatisfactory separation. C925 required an acidic medium range of 6–8 mol/L H_2_SO_4_ to separate Sc from Fe. In the TiO_2_ production, the final waste effluents typically contain about 2 mol/L sulfuric acid. As a result, C923 was proposed as a better extractant than commonly used (e.g., TBP and P204) for the separation of scandium from other metals such as iron and titanium [70].

With increasing solution concentration, the extraction of Sc increases, indicating a solvating extraction mechanism. Scandium can also be extracted from low-acidic solutions using neutral OPC extractants. For example, scandium extraction was significant under optimal conditions in low acidity salicylate media. Scandium can also be separated from Y(III) and La(III) depending on their pH sensitivity [82]. The selectivity of commonly used neutral phosphoric extractants (TBP, P350, C925, C923) is presented in Table 1.

### 2.4. Chelating Extractants

Acidic functional groups like –OH, =NOH, and –SH as well as coordination functional groups like =CO, =N^−^, and ≡N^−^ are found in chelating extractants. Several studies have been published that demonstrate the extraction of Sc, Y, La, and other transition metals using acidic β-diketone extractants such as HTTA and HPMBP [88,89,90]. It is commonly accepted that scandium extraction by an acidic β-diketone extractant (HA) proceeds through cation exchange and chelating processes. Based on Equation (6), the cation exchange mechanism could be pointed out. The formation of self-adduct species following the solvation by HA molecules via Equation (7) occurred in the presence of excess extractants [77,86].
(6)$${\mathrm{Sc}}^{3+}+3\bar{\mathrm{HA}}↔\bar{{\;\mathrm{ScA}}_{3}}+3H^{+}$$
(7)$${\mathrm{Sc}}^{3+}+4\bar{\mathrm{HA}}↔\bar{\;\mathrm{Sc}{\left(A\right)}_{3}\cdot \mathrm{HA}}+3H^{+}$$

Aqueous anions can also join the extracted complex through a method that is comparable to solvation, as previously stated. Researchers found that in weak acid solutions, scandium ions are coordinated with two extractant molecules 1-(2-ethylphenyl)-3-methyl-4,5-dihydro-1H-pyrazol-5-one (HA) and three perchlorate ions when extracted with chelating reagents and perchlorate, as shown in Equation (8) [91].
(8)$${\mathrm{Sc}}^{3+}+2\bar{\mathrm{HA}}+3{\mathrm{ClO}}_{4}^{-}↔\bar{\;\mathrm{Sc}{\left(\mathrm{HA}\right)}_{2}{\left({\mathrm{ClO}}_{4}\right)}_{3}}\;$$

With the use of HTTA in an aromatic solvent, scandium can be recovered quantitatively from solutions containing considerable amounts of iron and manganese. After reducing iron to the ferrous state, scandium can be recovered from the solutions in the pH range of 1.8–2.0 [88]. After that, a dilute mineral acid was used to strip it efficiently. According to Wand and Cheng [2], scandium in HCl and NaClO_4_ solutions was separated using 0.05–0.1 mol/L of N-antipyrinylformamide in chloroform, then recovered by an acidic aqueous solution of pH in the range 3.0 to 3.5, and finally precipitated by potassium hydroxide at pH 8.5. The precipitate was filtered, washed, and calcined to produce a 99.9% pure scandium.

### 2.5. Synergistic Systems

The most utilized compounds for synergistic solvent extraction of scandium are chelating and solvating extractants. It was discovered that the scandium extraction from hydrochloric solutions using a TBP-loaded solid support and a synergist with di-isooctyl methyl phosphonate (DIOMP) was greatly enhanced [89].

In comparison with TBP alone, the synergistic system significantly improved the separation factors for Sc over W, Al, and Y, increasing them from 20–60 to 600–1000. This resulted in the ability to extract scandium from diluted HCl solutions with less extractant than was previously achievable.

Wang and collaborators [55] postulated that the (trialkylphosphine oxide) TRPO impurity present in lower purity C302 plays a synergist role in scandium extraction based on the observation that un-purified C302 has a substantially better Sc extractability than purified C302.

Hirashima and collaborators [90] found that adding HTTA to P204 had a synergistic effect on the extraction of La(III) from a medium-containing chloride, whereas adding tributyl phosphate and acetyl acetone had an antagonistic impact.

Sudersanan [91] discovered that adding HTTA to P204 had a synergistic impact for extracting Ln from a medium containing chloride, whereas adding TBP, AA, or TOA had an antagonistic effect. Sudersanan also reported that Sc could synergistically be extracted by HTTA and P204, which was in agreement with the findings of [90]. Equation (9) shows the proposed extraction process:(9)$${\mathrm{Sc}}^{3+}+2({\bar{P204)}}_{2}+\bar{\mathrm{HTTA}\;}↔\;\bar{\mathrm{Sc}{\left(P204\cdot \mathrm{DEHP}\right)}_{2\;}\cdot {\left(\mathrm{TTA}\right)}_{2}}+3H^{+}$$

According to [92], a synergistic effect between the HPMBP and TBP was observed. The extraction of REEs presented elements other than Eu from 1 mol/L sodium perchlorate and perchloric acid by 0.2 mol/L of HTTA in carbon tetrachloride.

For Ln(III) extraction, HPMBP systems containing phosphine oxide, sulphoxide, and phosphine sulphide demonstrated synergism [93,94]. According to the literature, the use of HTTA or HPMBP in conjunction with the Aliquat 336 resulted in considerable synergistic effects in the extraction of specific trivalent lanthanides and actinides [95,96].

Even though there are numerous publications in the literature of synergistic systems for the extraction of Ln with chelating extractants, only limited information has been documented for scandium up to the present. A valid hypothesis is that the synergistic effects of the aforesaid systems on the extraction of scandium may be predicted based on the near chemical similarities between Sc(III) and Ln(III).

As a result, the SX of scandium in combination with synergistic systems is an interesting research task. A number of techniques concerning the synergistic extraction of scandium in association with systems comprising chelating extractants are presented in Table 2.

## 3. Processes for Scandium Recovery from Various Sources

The metallurgical processes for recovering scandium from different resources are reviewed in this section. Figure 5 gathers some of the scores, wastes, discharges, and disposal effluents. The objective is to select alternatives allowing scandium extraction, as a secondary element and put its recovery into the major flow-chart for the main metal extraction [1].

### 3.1. Scandium Recovery from Its Ores

Thortveitite and kolbeckite, two scandium minerals with high scandium content, are found in Madagascar and Norway’s thortveitite-rich pegmatites. Scandium and yttrium are found in Norway as a silicate called 2SiO_2_•Y_2_O_3_•Sc_2_O_3_ [97]. Figure 6 shows the chemical composition of another ore found in Madagascar with a Sc content of 42.6%, which looks to be a complex silicate of scandium, zirconium, and aluminum [1]. Sc can be recovered from the two kinds of thorveitite ores by fractional sublimation depending on the considerable difference in the sublimation points of their constituents. The powdered ore and carbon were roasted to 900–1000 °C in the presence of a stream of chlorine gas. Si, Ti, Al, Fe, and Zr chlorides were sublimated because their sublimation temperatures are less than 350 °C. At around 967 °C, ScCl_3_ was sublimated and deposited in a high-purity state in a zone where the temperature dropped to around 400 °C, whilst the YCl_3_ remained in the residue. In the United States, scandium was obtained from thortveitite-rich mine tailings, such as those from Darby, Montana [97]. As previously indicated, scandium-rich minerals are extremely rare. Only a few deposits able to be extensively mined have been discovered around the world. Therefore the recovery of scandium from other ores and waste products is very critical [100].

### 3.2. Scandium Recovery from Rare Earth Elements Ores

Scandium, like yttrium (Y) and lanthanides (La), is a rare earth element that can be found in nature [101]. Scandium concentrations range from 20 to 50 mg/kg in rare earth minerals such as monazite and bastnasite [15]. Bastnasite, the main mineral in the ore, contains Sc, Y, La, Fe, Th, and Ce, which accounts for nearly half of the REE mass in the Chinese ores.

The total extraction of the REEs, from the ore, including Sc, can be done by heating it in concentrated sulfuric acid at 250–300 °C and then leaching it using water. Th, Fe, Ca, F, and P are among the impurities co-leached into the liquid phase [102].

To extract scandium from a mixture of other metals, acidic OPCs such as P204, HEHEHP, and PC-88A can be utilized. With the use of HEHEHP as an extractant, sulfuric acid at concentrations ranging from 1–5 mol/L with ~20% co-extracted iron was suitable for extracting Sc from the solution [54]. With P204, Sc was extracted with over 90% efficiency from a leachate composed of 1 mol/L hydrochloric acid, leaving most of the Y and La in the solution [14]. However, due to the difficulty in stripping with acidic solutions, the applicability of these extractants is limited. There are trace concentrations of Sc present in the ionic-adsorption rare earth deposit (IARED) in China (about 10 mg/kg); however, these are readily leached from the deposit [61].

Digestion with solutions containing sulfate salts and precipitation by oxalic acid, as illustrated in Figure 7, concentrates the REEs in the IARED. In order to produce rare earth oxides, their concentrates are calcined. Scandium oxide and other rare earth oxides are dissolved together in HCl.

Additionally, scandium was separated from other REEs using two SX circuits developed by Liao and collaborators [71]. A sulfated kerosene solution containing naphthenic acid and iso-octanol was used for the extraction process. In this study, the scandium separation factors (β = Sc/REEs) were greater than 104, which confirmed that the scandium could be well separated from the other REEs. The Sc was enriched from 0.02–0.04% to 15–20% in rare earth oxides in the first circuit of SX, which used a cross-current protocol with 10 stages at an A/O phase ratio of 5:1.

In the second circuit, the obtained concentration of scandium was refined to high purity of 99.9% using a counter-current circuit with three extraction stages and three rounds of scrubbing. Scrubbing and stripping were carried out using two different solutions containing 0.35 mol/L and 1 mol/L of HCl, respectively. For the time being, China is the world’s leading exporter of rare earths, and some scandium is recovered during the rare earth-processing process. Since Sc and other REEs have chemical properties that are very similar, it is difficult to distinguish and separate between them.

To acquire high-purity scandium products, precise flow-charts procedures must be used, increasing the cost of both capital and operational equipment.

### 3.3. Scandium Recovery from Uranium Ores

Most uranium ores, including uraninite, contain traces of scandium. In 2020, the uranium production worldwide was 47,731 tons, from which 41% were produced in Kazakhstan, 13% in Australia, and 8% in Canada [103].

Consequently, separation of scandium as a by-product of uranium processing is of critical relevance in the nuclear industry. According to Lash [104], the uranium ores were ground, powdered, and leached with H_2_SO_4_. Then, it was found that the solution contained up to 1 mg/L Sc_2_O_3_. The uranium in the H_2_SO_4_ leachate solution can be entirely extracted using 0.1 mol/L of dodecyl phosphoric acid. However, it was impossible to recover the co-extracted Sc, Th, and Ti with U using HCl solution of 10 mol/L. By treating the 0.1 mol/L of dodecyl phosphoric acid with HF, both Sc and Th were precipitated. Subsequently, the accumulated Sc and Th could be recovered, as shown in Figure 8 [104].

The aqueous phase was filtered to remove the scandium and thorium fluoride precipitates, which contained 10% Sc_2_O_3_ and 20% ThO_2_, although the filtrate still included soluble titanium fluoride. Scandium hydroxide precipitate was formed by dissolving the precipitate in a 15% NaOH solution at 75–90 °C for 4 h. Following filtration, the Sc(OH)_3_ was dissolved with a specific concentration of HCl acid to ensure that impurities such as Ti, Zr, Fe, and Si were hydrolyzed. To separate Sc from the co-dissolved uranium and iron, it was precipitated with oxalic acid. The produced Sc_2_(C_2_O_4_)_3_ was thermally treated at 700 °C to yield an oxide product with a purity of 99.5%.

The preceding procedure is complicated by the fact that it involves many precipitation and dissolution steps. Additionally, the use of HF as a precipitation reagent may result in environmental contamination. Ross [24] proposed an SX technique for recovering scandium from uranium mill waste solutions using a primary amine such as Primene JMT as the extractant. Quantitative extraction of Sc was achieved in a small-scale pilot plant operation using 2.5% Primene JMT in kerosene at an A/O ratio of 50:1 from a waste solution that contained 0.6 mg/L of Sc.

Scandium was recovered from the organic phase using a 2 mol/L NaCl solution (pH of 1) at an A/O ratio of 1:10. Finally, an ammonia solution was used to precipitate the scandium. However, because of the risk of uranium radioactivity, the contamination of the environment is still possible.

### 3.4. Scandium Recovery from Ti and Zr Ores

Sc can be found in substantial concentrations in a variety of Ti minerals, including ilmenite (FeTiO_2_) and rutile (TiO_2_). For example, the magneto-vana-ilmenite ore (an ilmenite with a high vanadium content) in Panzhihua, China contains 0.002–0.004% Sc_2_O_3_ [105,106,107].

Titanium minerals are mostly found in Australia, Canada, India, Norway, and South Africa, with the rest of the world following closely after. The current global production of ilmenite is around 4.8 million tons per year, resulting in a possible yield of 96–194 tons of Sc_2_O_3_ from ilmenite [108].

Ilmenite is frequently processed further to make, among others, synthetic rutile and titaniferous slag. It was observed that the scandium was concentrated in the slag (e.g., 128 mg/kg [107]). Tailings had substantial levels of impurities in addition to scandium (40% SiO_2_, 16% FeO, 15% CaO, 14% MgO, 5.8% Al_2_O_3_, and 5.3% TiO_2_). The chlorination begins by fusing the Ti feed components together, resulting in vaporous TiCl_4_ [109].

Because the Sc is contained mostly in the residue or in the dust as scandium chloride, its recovery is feasible [110]. It has been stated that the dust includes a significant concentration of scandium, at least 132 ppm [108]. Figure 9 presents the flowchart of a process leading to ca. 90% recovery of Sc by treatment of ilmenite slag [111].

The process comprised the grinding of the slag, its roasting it with Na_2_CO_3_ at approximately 950 °C, and, finally, leaching it with 30% HCl at 80 °C and an S/L rate of 1:2. A total of 94% of the scandium was recovered from the leachate solution using kerosene containing 30% P204 at an A/O ratio of 20:1 and only 2.2% Fe co-extraction. Scrubbing with 5 mol/L HCl removed the extracted iron, while 2 mol/L NaOH recovered the scandium. After dissolving the scandium hydroxide precipitate in HCl, Sc was precipitated using oxalic acid. After heat treatment of the Sc_2_(C_2_O_4_)_3_ at 800 °C, a high purity of 99% of the Sc_2_O_3_ product was obtained. Feuling and et al. [112] obtained a patent for a procedure to recover Sc from a Ti chlorination residue using TBP as an extractant (Figure 10).

The radium was removed from the leachate solution by adsorption on fresh BaSO_4_ after the residue was leached with 6 mol/L HCl. TBP was used to separate scandium from primary elements such as sodium, calcium, and magnesium, as well as minor elements such as thorium, yttrium, and lanthanides in the filtrated solution. The co-extracted iron was left in the organic phase after the scandium stripping by 0.1 mol/L HCl [112].

Yagmurlu et al. [113] explained the potential of Sc recovery from a liquid containing iron chloride in the TiO_2_ production plants by the ScaVanger project. Additionally, the revolutionary ScaVanger processing method presents a comprehensive, viable, more environmentally friendly, and customizable process for generating Sc, Nb, and V compounds.

According to the flow-chart, ammonia was used to precipitate the scandium content as Sc(OH)_3_, which was then decomposed at 600 °C to obtain Sc_2_O_3_.

With a concentration of roughly 2 mol/L H_2_SO_4_, the hydrolytic solution from TiO_2_ production in the traditional sulfuric acid process contains 15–20 mg/L of Sc and other impurities, such as SiO_2_, Fe, Zr, Ti, and Lu [44]. The sulfuric acid solution contains Sc(III) ions, which confirm that the scandium can be extracted straight from the solution using SX. The authors investigated the extraction of scandium using C923 as a test method. Scandium was recovered with a high efficiency, while Ti, Fe, and Zr were extracted with low efficiency. In the scrubbing steps, acidic H_2_O_2_ or diluted H_2_SO_4_ were used to remove the co-extracted Ti. A scandium production with 96% purity and 95% recovery was achieved using 6–8 stages of SX, 7–9 stages of scrubbing, and 1–2 stages of recovery.

Scandium could be recovery from the mineral sands [114], which are processed for obtaining of zirconium. Usually, zircon sands are treated with HCl at around 1000 °C and, subsequently after zirconium separation, the scandium was found in the residual solution at a concentration of 0.34% [115]. Scandium can be extracted from zircon chlorination waste using a similar method like that used for the scandium recovery from Ti ores [112]. Scandium recovery from Ti and Zr residues is most typically accomplished using acidic or neutral OPCs as extractants. Unfortunately, because of the high levels of Ti and Zr in the residues, standard washing methods using diluted acids are unable to remove them. The co-extracted Ti and Zr can be eliminated with acidic hydrogen peroxide and fluoride, respectively, according to the literature [110].

### 3.5. Scandium Recovery from Tungsten and Tin Ores

Scandium can be found in tungsten/wolfram (W) minerals like wolframite (Fe–Mn-W) as FeWO_4_/MnWO_4_ and scheelite as CaWO_4_, which are mined and used to make around 37,400 tons of W concentrates each year. Tungsten ores are typically decomposed in an alkaline condition to produce an alkali tungstate solution, while Ca, Fe, and Mn are precipitate immobilized.

It was shown that scandium is present in substantial quantities in the residues from the treatment of tungsten ores [116,117]. As an example, scandium was found to be enriched to a level of around 0.04–0.06% in wolframite residues. Traces of scandium were also found in tin ores such as cassiterite (SnO_2_). A previous study discovered unusually high scandium content (ca. 1000 mg/kg) in wolframite and cassiterite in the Erzgebirge deposit [118].

A significant effort has been made to extract scandium from tungsten-containing materials. Due to the fact that scandium is predominantly present in the residue as Sc(OH)_3_, can be transformed to soluble salts via leaching by acids, such as H_2_SO_4_, HCl, and HNO_3_. Due to the production of stable Sc-Cl complexes, HCl can selectively leach scandium [119]. Scandium can be recovered from wolframite residues according to the flowchart presented in Figure 11, which also includes HCl leaching and P_2_O_4_ SX.

Concentrated HCl at 100 °C was used to dissolve scandium, obtaining a concentration of 95.3%. The feed solution had an acidity of approximately 2 mol/L of HCl and a scandium content of approximately 100 mg/L as Sc_2_O_3_. The scandium was extracted using P_2_O_4_ in kerosene at an A/O ratio of 1:4, yielding approximately 90% scandium. In the scrubbing stage, 3.5 mol/L of the HCl solution was used to remove the co-extracted elements, such as Fe, Ca, Mg, Al, REEs, and Si, from the organic phase.

In two stages, 2 mol/L of sodium hydroxide was used to completely recover the scandium. The resulting Sc(OH)_3_ had a Sc_2_O_3_ percentage of 70–78%, with a total recovery of 76–89%. Kim et al. and Gongyi et al. [119,120] indicated that about 100% of the scandium was extracted with the P204-impregnated resins from a leachate solution containing around 60 mg/L of Sc, 39 g/L of Fe, 19 g/L of Mn, and 0.40 g/L of W. The 2 mol/L hydrochloric acid can be used to remove the co-extracted Fe.

Wakui et al. [121] employed high-pressure leaching with strong HCl at 120 °C to dissolve a small amount of wolframite. The scandium in the filtrate was recovered using IX with an impregnated resin (PC-88A) after filtering the yellow-white tungstic acid precipitate.

After smelting cassiterite, scandium was extracted from a hydrochloric leachate containing 0.2 mg/L Sc, 0.4 mg/L Y, 68 mg/L Ca, 28 mg/L Fe, 27 mg/L Al, and 9.2 mg/L Nb. The main drawbacks of the hydrochloric-leach method are the hydrochloric evaporation and the creation of hazardous by-products, resulting in higher operating costs and contamination.

High scandium-leaching efficiencies are also achieved using the sulfuric acid-leach technique, but without the production of toxic chlorine gas. According to the literature [110], a wolframite ore comprising 0.04% scandium, 69.3% tungsten, 11.6% iron, and 4.8% manganese may be leached with strong sulfuric acid at elevated temperatures to obtain 94.9% Sc. By digesting tungsten, which contains about 23.7% iron, 22.5% manganese, 0.06% scandium, and 2% tungsten, with 18 mol/L of sulfuric acid for about 6 h at temperatures ranging between 100 and 140 °C, Vanderpool [122] demonstrated that nearly 100% of the scandium was leached. A carbon source was added to the solution in order to decrease the Mn content and then totally digest Sc, Mn, and Fe before the process was completed. Roughly 23.5 g/L of iron, 16 g/L of manganese, 0.15–0.23 g/L of tungsten, and around 0.04 g/L scandium were found in a leachate solution. Most of the W (84%) was left in the residual acid as H_2_WO_4_.

Rourke [123] described a method for recovering scandium from the waste of the tungsten production. For an operating time of 2 h, the wolframite residue was treated with 1 mol/L of sulfuric acid, including a 6% hydrogen peroxide solution at a S/L ratio of 1:25. After filtration of the slurry, the resulting solution had the following composition: 5.6 g/L of manganese, 3 g/L of iron, and 14 mg/L of scandium.

By using HTTA in toluene as a chelating extractant, scandium was totally extracted from the leachate in the pH range between 1.8 and 2.0. Sc(OH)_3_ or Sc_2_(C_2_O_4_)_3_ was precipitated by ammonia or oxalic acid after the scandium was recovered with 3 mol/L of hydrochloric acid. Zhong and collaborators [117] described a method for extracting scandium from W-slag sulfuric acid-leachate based on the presented diagram in Figure 12.

In order to remove thorium (Th), 0.2% of N1923 in kerosene at an A/O ratio of 4:1 was used to extract it from the leachate solution. This step was carried out after the reduction of ferric to ferrous ions by iron powder for removing the iron from the solution. An A/O ratio of 4:1 was used to recover more than 99% of the scandium from kerosene containing 4% N1923.

Furthermore, 3 mol/L of sulfuric acid and 3% hydrogen peroxide were used to scrub the co-extracted REE, iron, and titanium. Subsequently, the 2 mol/L hydrochloric acid was used for scandium recovery.

By precipitation via oxalic acid (C_2_H_2_O_4_) and thermal treatment of the Sc_2_(C_2_O_4_)_3_ precipitate, a Sc_2_O_3_ product with more than 90% purity and an 82% recovery rate was achieved. Scandium-rich wastes from the processing of tungsten and tin ores might be considered substantial sources of the metal.

Leaching with strong hydrochloric acid is possible; however, leaching by H_2_SO_4_ was preferred because of its lower pollution. Extraction and separation of Sc from Fe, Mn, and W in the leachate solution can be achieved using OPCs extractants. Scandium SX from H_2_SO_4_ solutions is easier with chelating and primary amine extractants than with HCl solutions.

### 3.6. Scandium Recovery from Nickel Ores

Scandium-rich nickel ores found in Australia are regarded as principal sources of the metal. It is estimated that the Ni and Co deposits in New South Wales with typical concentrations 130 to 370 mg/kg contain between 3500 and 6500 tons of scandium [3,21,124]. Scandium can be recovered as a by-product during the Ni and Co extraction.

Nickel (1–2%), cobalt (0.05–0.10%), iron (15–50%), Al (2–5%), and trace concentrations of scandium are all found in a standard nickel laterite ore (0.005–0.006%) from the above mentioned sources [16].

Using the high-pressure acid-leaching technique, scandium was readily leached from nickel laterite ores with H_2_SO_4_, yielding an impressive 94% recovery of the scandium [125]. When Fe and Al have been neutralized in the pH range between 2 and 4, Ni and Co can be recovered as sulphide precipitates, and scandium can be precipitated by increasing the pH above 4 [126]. It is noteworthy that the scandium is co-precipitated with a number of contaminants in this case. After the neutralization and precipitation processes, the scandium can be extracted from the impurities by using SX as an alternate separation method. For example, using acidic OPCs such as P204, PC-88A, and C272, significant scandium extraction was obtained from a solution with a pH range between 1.0 and 1.5, as shown in Figure 13 [19].

Following the Sc < Zn < Ca < Al < Cr < Mg < Ni extraction order, the selectivity was quite high. Several mining firms are now developing industrialized processes for the recovery of scandium from nickel laterites, which are projected to produce considerable quantities of scandium oxides in the future. One such example is a hydrometallurgy unit at the Nyngan project, which is expected to produce around 28,000 kg of Sc_2_O_3_ per year [127]. High-purity Sc_2_O_3_ production is expected to be accessible at a rate of 10,000–40,000 kg per year.

### 3.7. Scandium Recovery from Tantalum and Niobium Ores

The tantalum (Ta) mining industry in the United States also produced small quantities of scandium as a by-product of its mining operations [108,128]. Following leaching with water, a leachate containing scandium along with over 20 other metals in substantial quantities was produced from the tantalum sulphate tailings [129,130].

The results of the investigations showed that Sc could be entirely recovered from a solution of 50–200 g/L sulfuric acid using OPCs containing phosphoric and phosphonic acids, such as P204 and PC-88A. Although niobium (Nb), Ta, Y, and Fe were successfully separated, significant amounts of Zr, Hf, Ti, Th, and U were also co-extracted. To remove thorium from the organic solution, a 350 g/L sulfuric acid solution was used, followed by a 0.1 mol/L HF solution to remove other co-extracted elements. The recovery process from the loaded organic phase was carried out using 0.5–5 mol/L hydrogen fluoride solutions.

The observed order of recovery was Th > Ti > Zr > Hf > Sc > U > Sc. This resulted in the separation of Sc from Th, Ti, Zr, and Hf after 3–15 steps of counter-current back extraction as well as its recovery from uranium through six steps of fractional back extraction [126]. The most significant drawback of this procedure is the extensive use of HF acids, which poses a threat to the environment. It has been observed that the Nb resources in the Russian Arctic region include scandium oxide in unusually high concentrations ranging from 0.1 to 0.3% [131]. In some ores, niobium (7% Nb oxide) and REEs are found in combination with each other (10% of REE oxides). The rare earth minerals (such as Sc, Y, and La) were converted to hydroxides using a hydro-pyro-metallurgical treatment that included digestion with 45% NaOH [19]. In order to dissolve the scandium content in the hydroxide cake, HCl was used. A counter-current SX circuit using 80% TBP successfully recovered and separated Sc from the leachate, which contained rare earth, aluminum, and alkali-earth elements. Various hydrochloric acid concentrations can be used to eliminate the co-extracted Fe and U. Precipitation of the strip liquid with C_2_H_2_O_4_ leads to the formation of the Sc_2_(C_2_O_4_)_3_ precipitate, which was treated by heating to provide a 99.9% pure Sc_2_O_3_ product. The major improvement between this process and others is the separation of Sc from Nb and Ti via selective leaching and dissolution.

### 3.8. Recovery of Scandium from its Alloy Scraps

Scandium is wasted in large concentrations during the manufacturing of Sc-containing alloys like iron-scandium, aluminum-scandium, and magnesium-scandium due to its high reactivity with O, Cl, and F, particularly at high temperatures [16].

The high-Sc-Mg alloy, which is processed in a smelting process, is composed of 86% Mg and 14% Sc. Sc can be found in large concentrations in the salts containing wastes or metallic dross produced during the smelting process. The magnesium-scandium alloy dross had an average composition of 65–76% Mg, 13–22% Sc, and 1–1.6% Fe. After a one-stage leaching by hydrochloric acid and SX with P204, nearly 99% of scandium recovery and practically full separation from Mg could be obtained (Figure 14) [16].

During the SX stage, 99% magnesium and 90% iron were retained in the raffinate.

To remove the co-extracted Fe from the organic phase, HCl solutions with a low concentration can be utilized. To obtain the Sc(OH)_3_ precipitate, the scandium was recovered by 5 mol/L of NaOH. After thermal treatment, an oxide product was produced with 65.5% Sc, 0.6% Mg, and 0.4% Fe. Scandium can also be recovered from scraps of aluminum and iron alloys. The considerable amount of scandium present in the scraps formed during the Sc-alloy manufacturing cannot be considered as a resource worthy for valorization. Furthermore, there is currently no reliable information about this industrial processes suitable for this purpose.

## 4. Recovery of Scandium from the Bauxite Waste (Red Mud)

### 4.1. Acid Leaching

The bauxite waste (also named red mud (RM)) is frequently treated with acids to recover scandium and the other REEs [129,130,131,132,133,134]. Zhang et al. [132], using hydrochloric acid to extract scandium from red mud (HCl concentration: 6 mol/L; L/S-ratio: 4; temperature: 50 °C; leaching time: 1 h) obtained a Sc_2_O_3_ leaching rate of 82%. Wang et al. [133] also extracted scandium from the RM by HCl-leaching and found out that the most critical parameter was the L/S ratio, whereas the HCl-concentration had a significant effect only on the iron dissolution. The scandium leaching rate was higher than 85% using an HCl concentration of 6 mol/L, L/S ratio of 5, temperature of 60 °C, and reaction time of 1 h.

In addition, the HCl consumption during leaching was estimated to be ca. 21.2 mol/kg of RM. Scandium was recovered by Xu et al. [134] by leaching for 1.5 h at 80 °C using hydrochloric acid of 7 mol/L concentration and an L/S ratio of 8. In another study, Tang et al. [135] observed that the leaching of scandium from RM of particle size of 65–80 µm using sulfuric acid of 50% concentration, an L/S ratio of 3, and a temperature of 90 °C and duration of 3 h was above 85%. A series of leaching studies using different acids such as hydrochloric, nitric, sulfuric, acetic, methanesulfonic, and citric acids were also carried out on Greek RM [136]. In the mentioned study, the effect of several parameters, including variation of acid concentrations, L/S ratio, leaching time, and temperature, were investigated. The scandium content of the utilized Greek RM was approximately (ca. 120 mg/kg). The results of this work indicated that the acid was utilized during the initial leaching stage for increasing the pH, neutralizing the RM alkalinity, and dissolving the aluminosilicates. The percentage of REEs extracted improved with the increase of the acid concentration, L/S ratio, and the leaching duration. For mineral acids of 0.5 mol/L concentration, the leaching temperature had little effect on extraction, while for citric acid leaching, the extraction of REE increased with the increase of the temperature.

Hydrochloric acid leaching showed the highest REE extraction rate compared with other acids. Because the different ionic radii of the REE from minerals, they do not show similar leaching rates in specific environments [30,137,138].

At the following leaching conditions: 24 h of leaching time at room temperature with a 6 mol/L hydrochloric acid concentration and an L/S ratio of 50, the extraction rate of neodymium, dysprosium, and yttrium was found to be greater than 80%, while this of lanthanum, cerium, and scandium was from 70% to 80%. During a digestion time of 24 h at room temperature and an L/S ratio of 50, the dissolution of iron increases with increasing acid concentration, from 5% at 0.5 mol/L to about 60% at 6 mol/L hydrochloric acid.

The scandium extraction was found to show a very close relationship with Fe dissolution in RM. About 50% of scandium was recoverable from RM with just minor Fe dissolution. On the other hand, a small increase in the scandium extraction above 50% resulted in enhanced Fe concentration in the solution [133].

Scandium was found in individual minerals or in the outer layer of iron oxide particles in the RM because it was not evenly distributed throughout the iron oxide phases. Liu and collaborators [139] observed that RM was constituted of hematite, goethite, quartz calcite, anatase, zeolite and gibbsite as the mineral phases of the red mud. They also employed time of flight secondary ion mass spectrometry (TOF-SIMS) and electron-probe micro analysis (EPMA) to check into the affinities of Sc and Ga with the primary elements found in red mud, which included Fe, Al, Si, Ti, and Ca.

Scandium leaching from RM was assumed to be primarily regulated by diffusion from the outermost surface of the iron particles. They also discovered that hydrochloric acid concentrations less than 3 mol/L were favorable for maximum scandium leaching and less Fe dissolution into the solution. Because Fe is easily dissolute in the acid digestion and the hematite (Fe_2_O_3_) is a major chemical component of RM, different hydrometallurgical procedures were developed for extraction of Sc and other valuable or precious metals from RM [18].

For the pretreatment, a reduction-sintering method was used, in which RM was mixed with a carbon source, CaCO_3_, and Na_2_CO_3_ at 800–1000 °C. After that, a brown mud was obtained by re-leaching alumina with hot water at 65 °C [21,25]. The brown mud after reduction included 90% reduced iron as Fe_3_O_4_ and 99% from this product yield for pig iron. Scandium (420 mg/kg) was found in the residue as well as TiO_2_ (19.4 wt%), Ln (1.47g/kg), and Y (180 mg/kg) [19]. Afterwards, the pig iron was separated from the residue, which was dissolved with sulfuric acid, and the Ti was recovered by hydrolyzing the leaching solution at 140 °C [18].

Following the recovery of Ti, the Sc can be extracted further by SX or precipitation from the solution depending on the conditions. Ochsenkühn et al. [140] extracted Sc, La, and Y from RM by selective leaching with dilute nitric acid under normal conditions without any pretreatments. Their findings revealed that the best Sc (80%) and Y (96%) recovery was obtained by using 0.5 mol/L nitric acid leaching for 24 h, a L/S ratio of 50, at room temperature.

Moreover, they found that scandium has the highest leaching selectivity over Fe under these conditions with only 3% dissolution of iron. This means that the pretreatments steps of the RM such as screening, magnetic separation, and thermal treatment were unnecessary, because most of the major RM components, particularly iron, were not significantly dissolved by this selective leaching method.

The dilute nitric acid leaching approach was subsequently scaled up to a pilot plant running at room temperature and pressure, as illustrated in the proposed flow-chart in Figure 15 [140].

For the RM, containing high-hematite (Fe_2_O_3_) concentrations (>30 wt%), the carbo-thermal reduction pretreatment is appropriate to recover or eliminate the iron [141].

Sulfuric acid can be employed as a digestion agent to dissolve the RM prior to the scandium recovery because of its low cost and environmental impact. Due to the suitable leaching efficiency of scandium, it is believed that direct acid leaching with dilute nitric acid is acceptable for RM with a low iron oxide content (less than 30 wt%) [138]. Recently Boyarintsev et al. [142] evaluated the leaching of scandium from the Russian RM using 2.0 mol/L Na_2_CO_3_ under carbonation conditions (CO_2_ gas) in the pH range of 9.5–10.0.

The obtained results indicated that the concentration of Sc is highly variable and depends on the ultrasonic treatment time, the Na_2_CO_3_ concentration, the temperature, and the pH value. In the absence of CO_2_ gas, the maximum Sc concentration in one leaching step equivalent to 20% was found when 1 mol/L NaHCO_3_ solution was used at 70 °C for 120 min, compared with 8.1% for 1 mol/L Na_2_CO_3_ solution at 70 °C and 17.4% for 2 mol/L Na_2_CO_3_ solution at 90 °C.

Bioleaching was investigated by Kiskira et al. [143] as a new and environmentally compatible method for scandium extraction from RM. Different parameters were studied using different microbial cultures on Greek RM. They observed that the maximum scandium extraction was 42% when using acetobacter tropicalis in a one-step bioleaching process at 1% S/L. Moreover, the main organic acids produced were acetic, oxalic, and citric.

### 4.2. Solvent Extraction

SX is a frequently utilized process for scandium recovery from acidic solutions. OPCs such as D2EHPA/HDEHP/P204, TBP, HEHEHP/Ionquest 801/P507, and C272 are mostly used for this purpose. Several studies investigated the extraction of scandium from leachate solutions of RM using the above extractants [2].

A general description of the scandium recovery procedure from RM leachate is presented in Figure 16.

In this context, Xu et al. [134] used 8 vol% of P204 for the scandium extraction from real RM leachate solutions (HCl), achieving a 97.9% efficiency under the following conditions: A/O phase ratio of 10, extraction duration of 2.5 min, and concentration of the alcohol (ethanol) additive 4 vol.%. Zhang et al. [144] used 1 vol% of P507 as extractant in kerosene to recover scandium from RM leachate HCl solution (scandium content of 6.7 mg/L). Consequently, with an O/A phase ratio of 1 and extraction time of 15 min, the scandium extraction efficiency was greater than 90%. Following the SX, the organic layer was scrubbed twice with 6 mol/L of HCl and distilled water, respectively, at an O/A phase ratio of 3. Moreover, the recovery process was carried out under the following conditions: 2 mol/L NaOH, 15 min contact time, O/A phase ratio of 3, and temperature of 50 °C. Following that, the Sc(OH)_3_ was additionally dissolved by 6 mol/L HCl and the pH of the produced solution was adjusted to 1.5 using ammonium hydroxide prior to the precipitation step. Finally, a material enriched in Sc_2_O_3_ with a purity of 66 wt% was achieved through the use of oxalic acid precipitation followed by thermal decomposition at ~800 °C.

Liu et al. [145] evaluated the extraction performance of Sc from the synthetic leachate of sulfation-roasted RM using OPCs (P204, P507, and V10). As a result, they observed that P204 has a high extraction ability toward Sc. Furthermore, P204 can be directly applied to the real leachate of H_2_SO_4_-roasted RM with a low concentration of Fe^3+^ and Si^4+^, and 96.5% Sc can be recovered from the organic phase (P204/sulfonated kerosene).

Jiang et al. [146] performed comparable investigations where they extracted scandium from a hydrochloric acid RM leachate solution with a scandium concentration of ~300 mg/kg using as extractant kerosene containing 3 vol% P204 and 5 vol% naphthenic acid.

The purity of scandium oxide was around 50 wt%. In another study [147], a solution of 6 mol/L hydrochloric acid was used to dissolve the RM for 4 h at 60 °C using a solid/liquid volume ratio of 4. After extraction, the following composition was obtained: 8 mg/L of scandium, 9 mg/L of sodium, 7 g/L of iron, 12 g/L of calcium, 15 g/L of aluminum, and 3 g/L of titanium.

Scandium was subsequently extracted by adsorption on TBP-modified active carbon. However, due to the co-adsorption of titanium, low adsorption efficiency of scandium was observed. Wang and Li [145,148] mentioned that by using an emulsion LM carried by C272, it was possible to extract scandium from the RM leachate solution.

Based on the acid leaching and solvent extraction procedure, Li and collaborators explored the feasibility and mechanism of recovering Sc from a Sc-rich material. They observed that H_3_PO_4_ outperformed HCl, H_2_SO_4_, and HNO_3_ in separating Sc from impurities [30]. It was also shown that if H_3_PO_4_ concentration was 6–8 mol/L and the leaching temperature was 120–140 °C, the leaching period was 60–90 min and the L/S ratio was 10–12 mL/g, the leaching of Sc approached 90%.

Wang et al. [46] performed SX experiments for scandium recovery from a synthetic solution based on Australian RM. This solution which contained 5.53 mg/L scandium was prepared by mixing the sulfuric acid (1 mol/L) with the Australian RM at an L/S ratio of 10, heating it to 50 °C, and allowing it to leach for 2 h.

Different extractants, such as an acidic OPC (P204, Ionquest 801 and C272), a primary amine (Primene JMT), a carboxylic acid (V 10), and two chelating materials (LIX 984 N and LIX54-100), were investigated to separate scandium from other metals in the synthetic leach solution. Scandium was extracted using P204, Ionquest 801, C272, and Primene JMT with an efficiency of over 99%, as shown in Figure 17 [46]. On the other hand, the mentioned OPC (P204, Ionquest 801, and Primene JMT) presented high Fe co-extraction rates, being even higher than 54%.

Botelho et al. look towards recovering scandium and zirconium from RM after being leached with 20% H_2_SO_4_. Separation studies were conducted out utilizing two C923, D2EHPA, Alamine 336, and TBP, and their mixes. It was found that at all pH values, C923 and Alamine 336 were more selective for Zr than for Sc. At pH 1.5 and 2.0, D2EHPA was more selective for scandium extracting 89% and 40% of Zr, respectively as shown in Figure 18 which explains the practical steps of the recovery process [150].

As a phase modifier, TBP was added to these three OPCs extractants (P204, Ionquest 801, C272) in order to increase their first phase separation ability, which was unsatisfactory when used alone, i.e., only one single component.

The organic system contained Shellsol D70 as a diluent, 0.05 mol/L P204, and 0.05 mol/L TBP at an A/O ratio of 5. The temperature of the system was 40 °C, and the pH of the aqueous phase 0.25. The above operational conditions were determined to be the most effective for the scandium extraction. It was possible to extract almost 99% of the scandium under these conditions, where Zr, Ti and V were also added.

The precipitated Sc(OH)_3_ was obtained after the P204/TBP system was washed twice with water and subsequently back extracted from the P204/TBP with NaOH.

Wang and collaborators [57] developed a potential flow-chart for the scandium recovery from the H_2_SO_4_ RM leachate solution. According to their findings, P204 can commonly be used for the scandium extraction from RM solutions due to its efficiency and capacity. Furthermore, a higher scandium recovery rate can be obtained by combining P204 with TBP. Although acidic OPCs can achieve high scandium extraction from leachate solutions, impurity metals, particularly Fe, may be co-extracted depending on the leaching conditions.

As a result, it is critical to investigate the best SX conditions in order to maximize the scandium recovery from RM while minimizing other metals. In another study, the selective extraction of scandium from the Greek RM was studied by Ochsenkühn et al. [151]. The flow chart of the procedures is shown in Figure 19.

The dried RM powder was sintered with sodium potassium carbonate and sodium tetraborate at 1100 °C for 20 min, dissolved in concentrated hydrochloric acid (1:1), and then diluted with 1.5 mol/L hydrochloric acid. A chromatographic column (IX) loaded with Dowex 50 W-X8 cation exchanger pre-treated with 2 mol/L hydrochloric acid was used to pass the leachate solution. The IX was followed by 1.75 mol/L hydrochloric acid elution of impurity metals such as Fe, Al, Ca, Si, Ti, and Na and minor elements like Ni, Cr, Mn, and V followed by 6 mol/L hydrochloric acid sequential elution of Sc, Y, and Ln.

The Sc, Y, and Ln eluate solution was adjusted to a pH value around zero with NH_4_OH before SX using 0.05 mol/L of P204 in hexane at an A/O ratio of 10. An extraction efficiency of 99% was obtained, which resulted in the scandium extraction into the organic phase, while most of the other elements were allowed to remain in the aqueous one.

Scandium was then recovered from the strip liquid and purified as mixture of [Sc(OH)_6_]^3−^ and Sc(OH)_3_ using 2 mol/L NaOH at an O/A ratio of 1:1.

The recovered scandium solution obtained would next be diluted with hydrochloric acid before being precipitated with oxalic acid. Following that, pure Sc_2_O_3_ could be produced by calcining Sc_2_(C_2_O_4_)_3_. For the overall process, 94% of the scandium was recovered from the RM using a sequence of roasting, acid digestion, IX, SX, and back extraction.

Wang et al. [57] investigated a selective acid leaching combination with SX of scandium from an Australian RM and suggested a conceptual flow-chart for scandium recovery, which is shown in Figure 20. SX with Primene JMT (first stage) was used to separate Zr and Ti from the RM, whereas the second stage was carried out 0.05 mol/L P204 and 0.05 mol/L TBP for scandium recovery from the leachate solution (1 mol/L H_2_SO_4_). Scandium was recovered into a liquid phase of [Sc(OH)_6_]^3−^, which was then diluted with water to precipitate scandium as Sc(OH)_3_. To achieve pure Sc_2_O_3_, additional purification processes such as the dissolution of Sc(OH)_3_ with hydrochloric or sulfuric acid, precipitation as scandium oxalate, and calcination could be used.

Shoppert et al. [152] investigated the feasibility of selective Sc leaching from RM using a diluted acid (pH > 4) with MgSO_4_. In this context, different parameters such as temperature, time, pH, and MgSO_4_ concentration were tested to see how they affected the scandium extraction efficiency. They observed that Sc can be extracted in excess of 63% at pH 4, 80 °C, after 1 h, whereas a higher content could be extracted at pH 2. Furthermore, increasing the pH from 2 to 4 reduced the iron extraction from 7.7% to 0.03%.

Pasechnik et al. [153] developed a method for selective scandium extraction by removing silicon by sulfatization, during the first stage of scandium concentrate dissolving. They observed that increasing the acidity promotes the precipitation of NH_4_Sc(SO_4_)_2_, which effectively separates scandium from impurities such as Al, Fe, Zr, Ti, and Ca as shown in Figure 20.

Recently, Shoppert et al. [154] studied the use of weak nitric acid (pH 2–3.5) to selectively leach Sc from the red mud generated by the alkali fusion-leaching process. They observed that pH has a substantial impact on the degree of Sc extraction. When the pH is reduced from 3.5 to 2, the Sc extraction after 90 min of leaching at T = 70 °C increases from 10% to 68.5%, and the iron extraction increases from 1.3% to nearly 5%. Lei et al. [155] devised a way to extract scandium and titanium from red mud leachate solution using a neutralization precipitation and acid leaching methodology as shown in Figure 21. They found that the precipitation efficiencies of scandium and titanium were 93.74% and 99.47%, respectively, under optimal conditions. Furthermore, according to their results, scandium and titanium lost 6.3% and 5.9% of their value, respectively.

In order to achieve an environmentally friendly separation, Jumadilov et al. [156] investigated the effect of Nd(III) and Sc(III) ionic radii on their sorption process from aqueous phase using individual ion exchangers such as Amberlite IR120, AB-17-8, and an interpolymer system Amberlite IR120-AB-17-8. They reported that the initial cation and anion exchangers’ highly ionized states can be expected during their remote contact in interpolymer systems.

This enables the development of novel sorption technologies and processes. Table 3 summarizes the most recent studies and research on the scandium recovery from the RM.

An interesting study by Rout et al. [163] looked into how hollow fiber-supported liquid membranes (HFSLM) and SX techniques could be used to extract Sc from a mixture of metals that included Cu, Ni, Zn, Fe, Co and Mn. They compared both strategies which could be valuable for industries, while Cyanex 272 was used as a mobile carrier. During their study, they investigated different parameters such as pH, Cyanex 272 concentration, flow rate, metal ion concentration, and sulfuric acid concentration as striping agent. They found that Sc can successfully separate from the multicomponent metallic solutions (Cu, Ni, Zn, Fe, Co and Mn) under the following conditions: pH 2.0, 0.3 mol/L Cyanex 272 in membrane phase, 240 mL/min flow rate, and 1.0 mol/L sulfuric acid in the strip solution.

## 5. Limitations, Challenges and Future Research Directions for Scandium Recovery

Scandium availability is limited because of its natural deficiency and high production costs in Asia and Russia, while it is not produced in Europe. As a result, the extraction of scandium from other resources, such as secondary resources in Europe, is a significant cause of concern. Among the most crucial scandium recovery challenges, the co-dissolution and co-extraction of metal impurities such as iron, titanium, zirconium, uranium, thorium and rare earth, which interfere with the scandium extraction [10,45], are mentioned. Due to the presence of several alkaline materials, the hydrometallurgical operations are typically characterized by significant acid consumption since the pH of the residue is exceptionally high [164]. As a result, some of the acids must be used to neutralize the alkaline products left over from the after the digestion process. While a high acid concentration can result in a significant recovery of REEs, it can also reduce the efficiency of the process because other elements can be dissolved. It is important to note that the presence of significant elements in the leach solution, particularly Fe(III), Al(III), Ti(III) and Zr(III), is a significant drawback for the separation process (solvent extraction). These elements are difficult to separate from REEs due to their similar trivalent oxidation state and ability to be extracted concurrently in most separation processes [165].

This work explored the most recent applied protocols for scandium recovery by solvent extraction technique using different extractants from various sources. We reviewed the most important previous published papers related to the recovery of scandium from various sources (ores and waste) for the period from 1969 to 2022. We hope that this review could be useful for all which are interested in scandium recovery at industrial scale from all existing scandium sources, providing a solid information background for such applications.

## 6. Conclusions

To separate and purify scandium from a variety of solutions, solvent extraction is the most extensively applied technique. In addition to being found in ores of other metals, scandium can also be recovered from the wastes and tailings of other metals. When the comparatively high scandium content and availability are taken into consideration, red mud can be considered a valuable and potential scandium resource rather than as a solid waste. Scandium recovery from red mud should be considered, because of its rarity and increasing market demand, as a primary goal rather than a secondary by-product in the design process of the entire flow-chart. One of the most popular approaches for scandium recovery from red mud applies hydrometallurgical processes, such as acid digestion, SX, and IX. Industrial-scale operations for the extraction of scandium from bauxite wastes and nickel laterites are currently being considered. Due to their high energy consumption, pyrometallurgical techniques are not appropriate for scandium recovery from wastes and sludge. The most extensively used processes for the scandium recovery are hydrometallurgical. Contaminants in high concentrations could be removed, regardless of the cost increase, using strong mineral acids during the digestion process.

Several unwanted metals, such as Fe, Al, Zr and REEs, could interfere with the scandium extraction if they were simultaneously dissolved together or extracted at the same time. Some of OPCs extractants have several disadvantages, the most notable of which are the difficulty in recovering of Sc and the insufficiency of selectivity.

In order to improve the extraction selectivity and efficiency, new techniques or optimizations to current processes must be developed in the area of digestion methods and SX. Red mud and nickel laterite ores are identified as the most interesting and promising scandium resources for future valorization.

Based on our findings during scandium recovery from red mud, the hydrometallurgical methods (acid leaching and solvent extraction) were effective. Moreover, the organo-phosphorus compounds (TBP and D2EHPA) showed interesting potential taking into consideration some co-extracted metals such as Fe(III) and Ti(III). In the case of iron, it was found that the removal of iron content before scandium recovery is the optimum choice, by using solvent extraction with trinoctylamine (N_235_) or diethyl ether, from the red mud leachate (8 mol/L HCl). The solvent extraction by diethyl ether was preferred due to its high efficiency and better economy. In order to remove the Ti(III) content from the leachate, the scrubbing process by 3 mol/L of HCl or H_2_SO_4_ was proposed, depending on the leachate.

## Figures and Tables

**Figure 1 materials-15-02376-f001:**
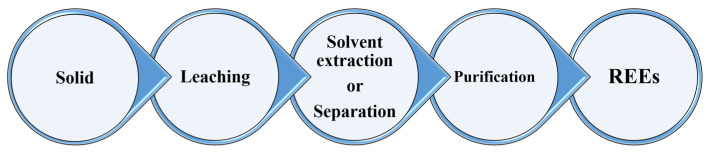
Representation of separation and purification procedures of rare earth elements.

**Figure 3 materials-15-02376-f003:**
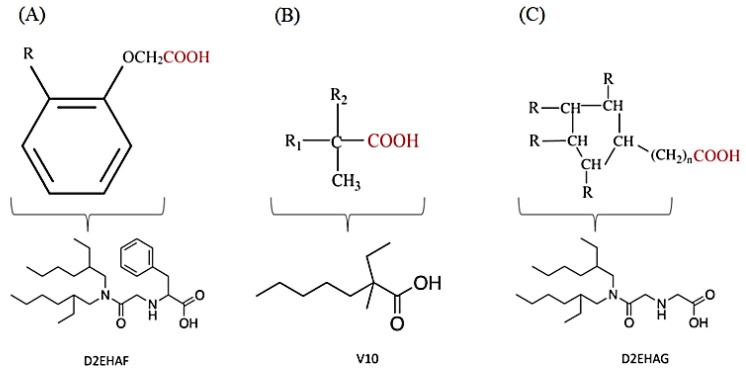
Chemical structure of the carboxylic acids used as typical scandium extractants: (**A**)—phenoxy acetic acid, (**B**)—versatic acid, and (**C**)—naphthenic acid, adapted from [2,69].

**Figure 4 materials-15-02376-f004:**
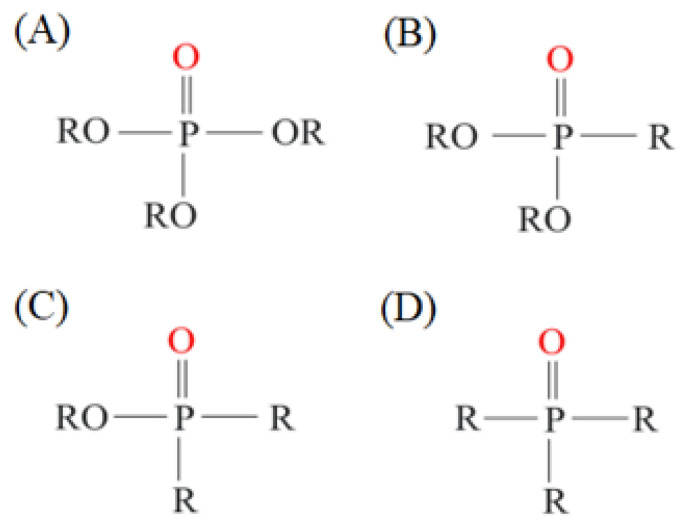
Chemical structure of solvating extractants based on phosphorus: (**A**)—phosphate, (**B**)—phosphonate, (**C**)—phosphinate, and (**D**)—phosphine oxide, adapted from [2,9].

**Figure 5 materials-15-02376-f005:**
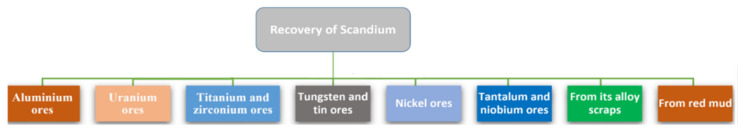
Scandium recovery from different sources.

**Figure 6 materials-15-02376-f006:**
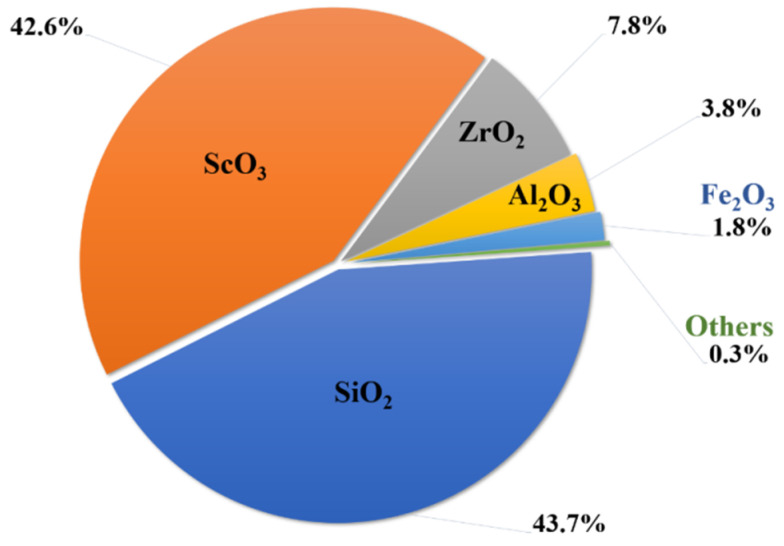
The elements contained in the scandium ore from Madagascar, adapted from [1].

**Figure 7 materials-15-02376-f007:**
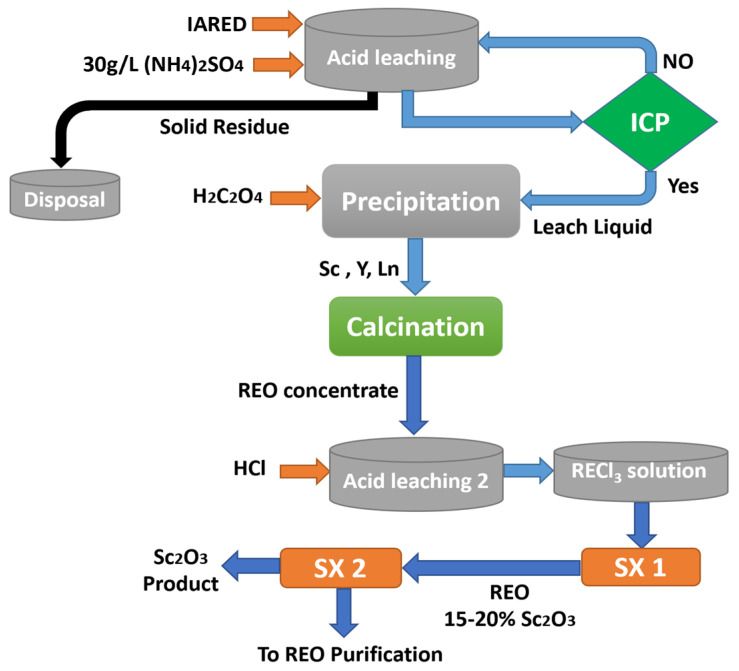
A flow-chart for scandium recovery from IARED, adapted from [71].

**Figure 8 materials-15-02376-f008:**
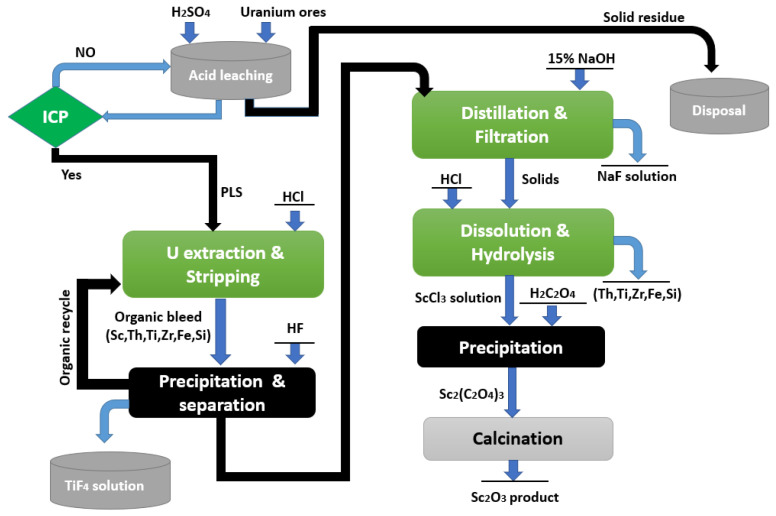
Scandium recovery from uranium ores, adapted from [104].

**Figure 9 materials-15-02376-f009:**
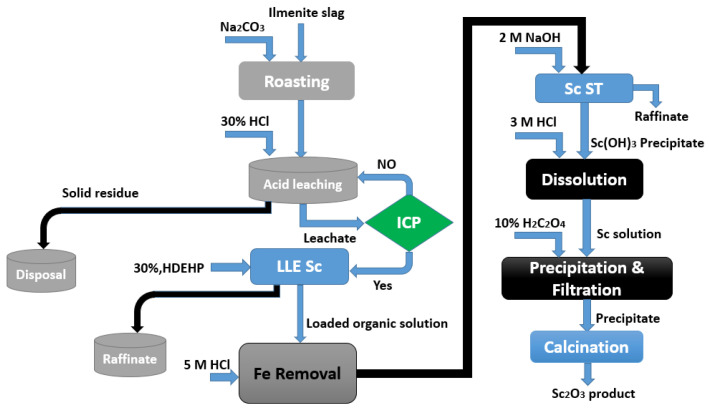
The process of scandium recovery from the ilmenite slag, adapted from [111].

**Figure 10 materials-15-02376-f010:**
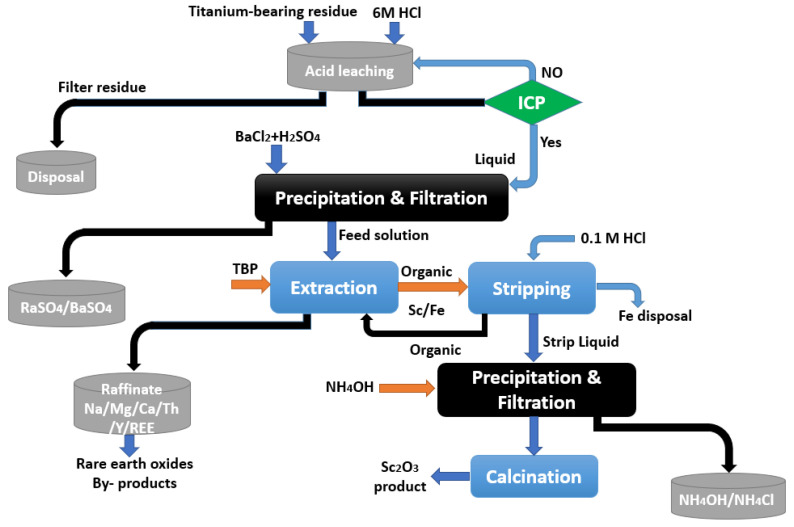
Flow-chart for scandium recovery from titanium-bearing wastes, adapted from [112].

**Figure 11 materials-15-02376-f011:**
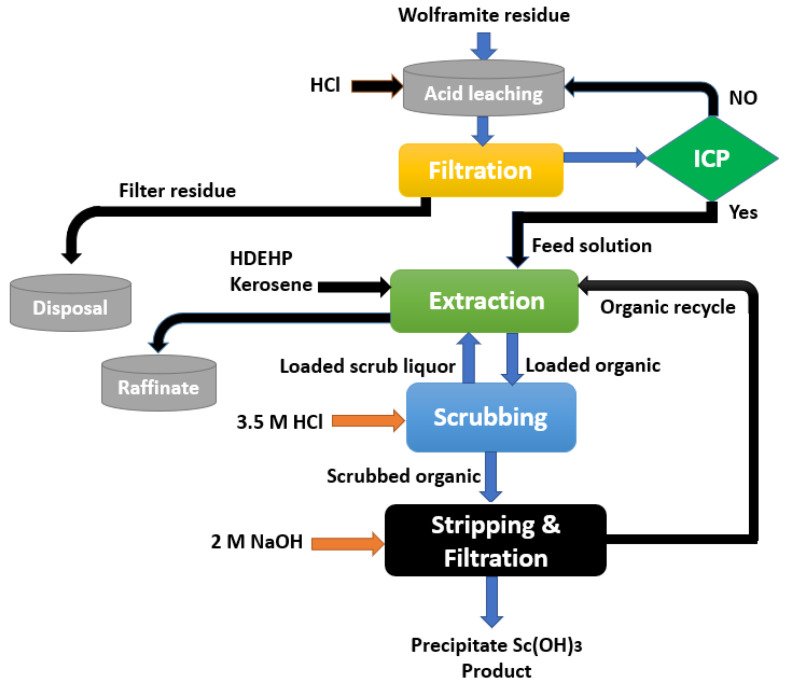
Scandium recovery from wolframite residue, adapted from [119].

**Figure 12 materials-15-02376-f012:**
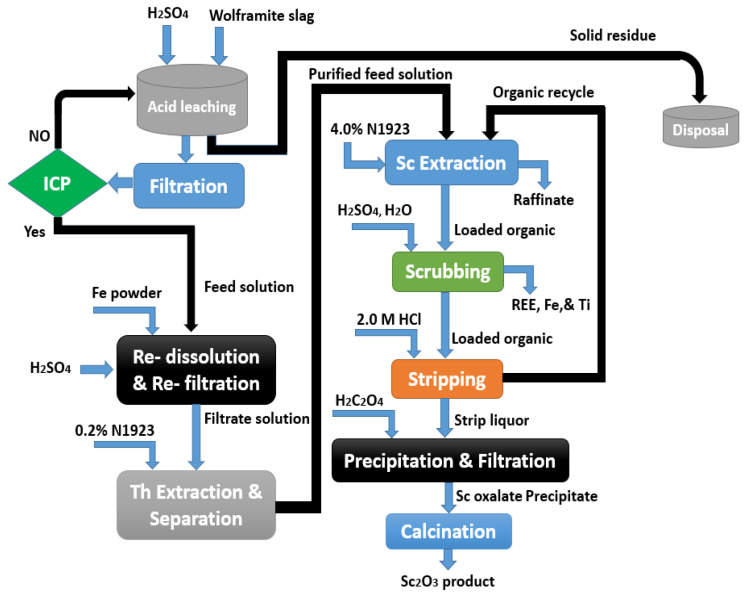
Scandium recovery from tungsten slag, adapted from [117].

**Figure 13 materials-15-02376-f013:**
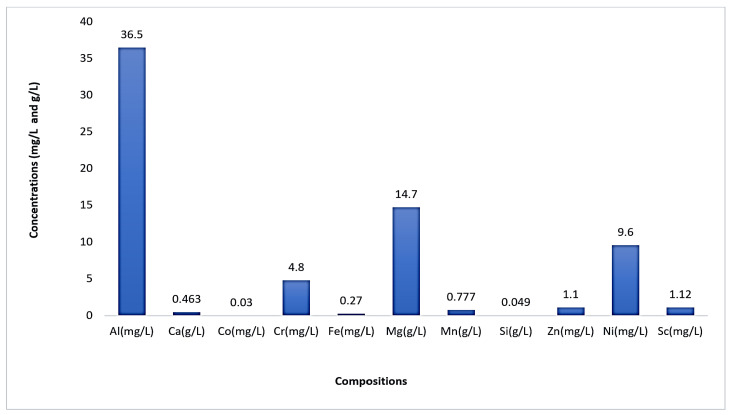
The composition of laterite ore H_2_SO_4_-leached solution, after neutralization and sulphide precipitation, adapted from [19].

**Figure 14 materials-15-02376-f014:**
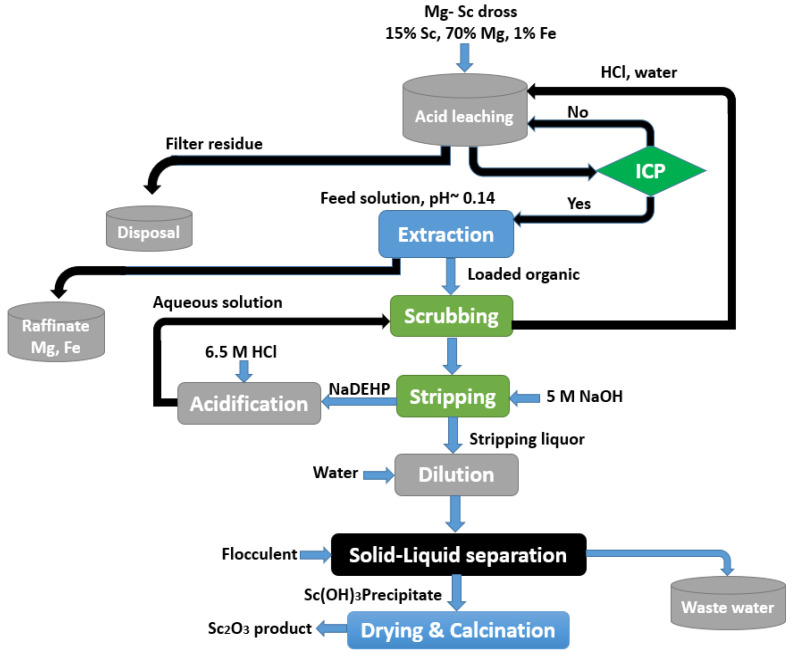
Scandium recovery from Mg–Sc dross, adapted from [16].

**Figure 15 materials-15-02376-f015:**
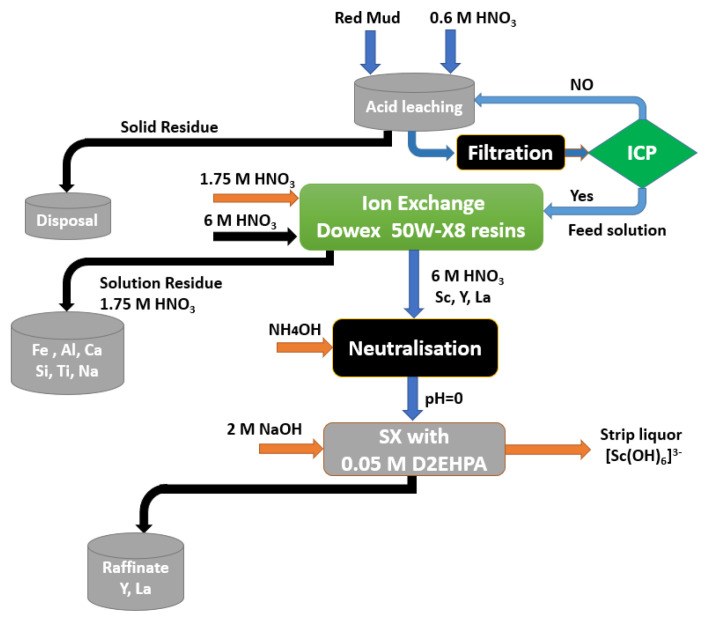
Proposed flow-chart for scandium recovery from RM, adapted from [140].

**Figure 16 materials-15-02376-f016:**
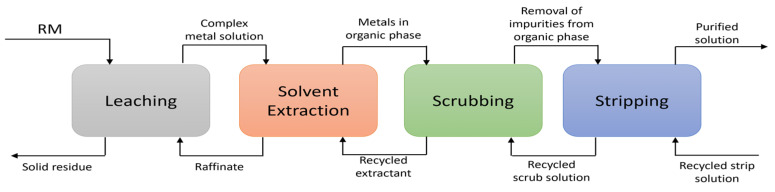
General process for scandium recovery from RM leachate.

**Figure 17 materials-15-02376-f017:**
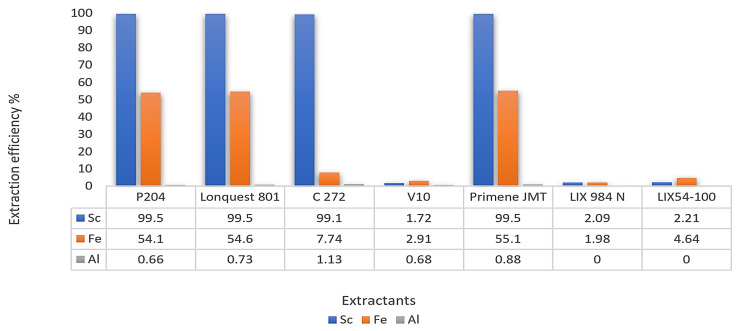
Extraction efficiency for different extractants at a pH of 1.5–2. (Shellsol D70 used as a diluent containing 3% of the extractants at an A/O ratio of 5), adapted from [149].

**Figure 18 materials-15-02376-f018:**
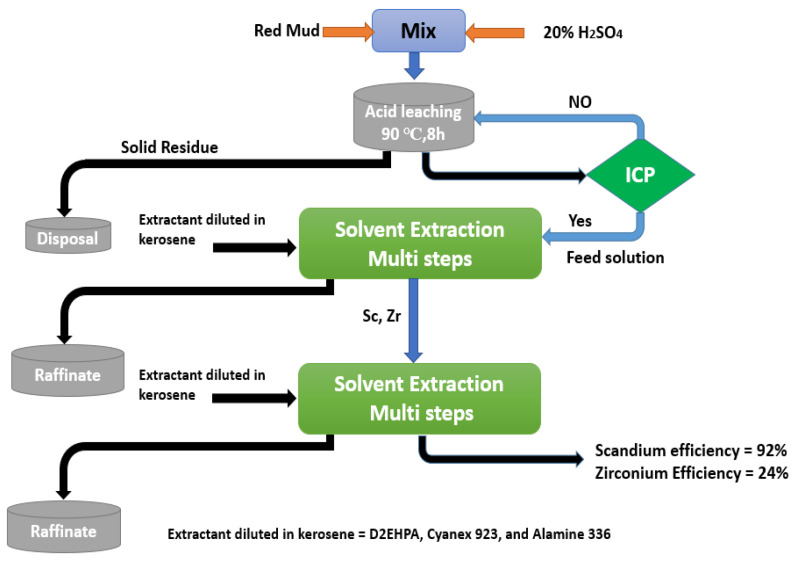
Flowchart proposed for Sc and Zr recovery from RM by SX, adapted from [150].

**Figure 19 materials-15-02376-f019:**
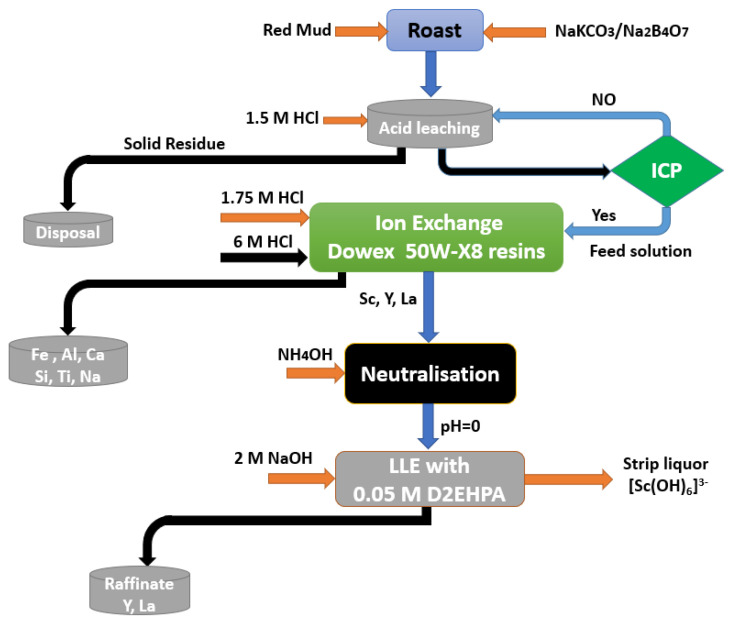
The proposed flow-chart of scandium separation from RM, adapted from [1,151].

**Figure 20 materials-15-02376-f020:**
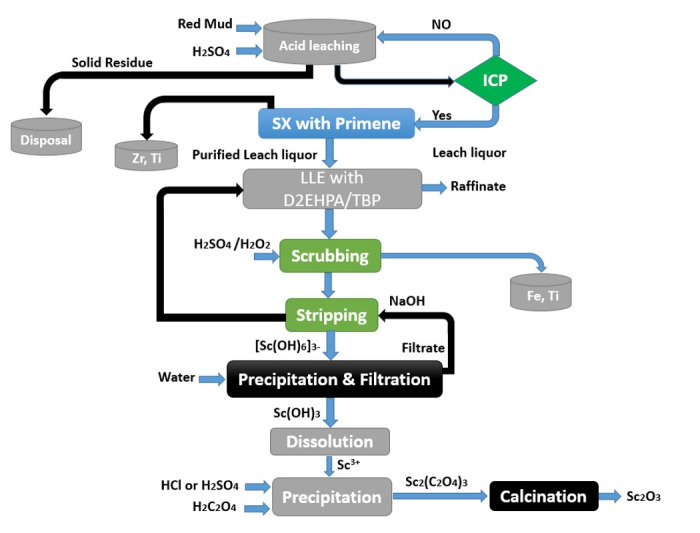
The proposed flow-chart of scandium separation from Australian RM, adapted from [57].

**Figure 21 materials-15-02376-f021:**
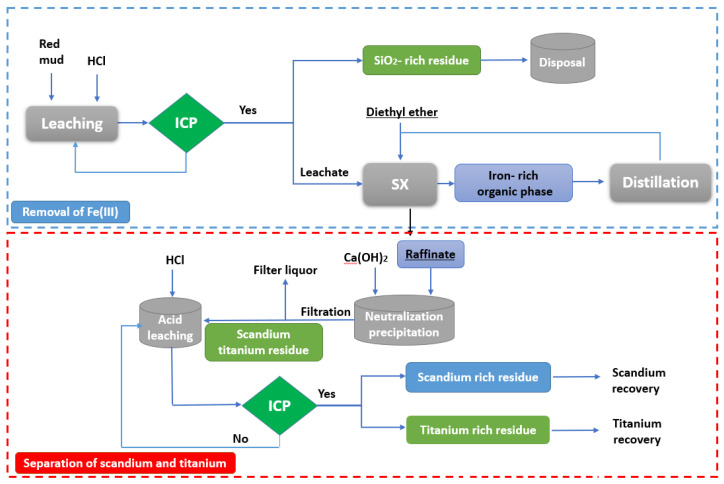
The proposed flow-chart of scandium separation from Guangxi Pinglu RM, adapted from [155].

**Table 1 materials-15-02376-t001:** Overview of studies on scandium extraction using the SX method.

No.	Author(s)	Extractant Structures	Metals Matrix	Remarks	Ref.
1	Qureshi et al. (1969)	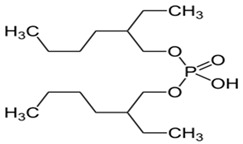 HDEHP, D2EHPA, P204	Sc^3+^ ∼ Ti^4+^, Zr^4+^,Hf^4+^ > Y^3+^ > La^3+^ > Mn^2+^	The organic is 0.75 M of HDEHP in n-heptane or cyclohexane the aqueous feed is 1–11 M HCl, HClO_4_ or HNO_3_	[61]
2	Xue LZ et al. (1992)		Sc^3+^ >> Fe^3+^ > Lu^3+^ > Yb^3+^ > Er^3+^ > Y^3+^ > Ho^3+^	The organic phase included P204 in n-octane. The aqueous phase pH 3–10 M HCl	[75]
3	Ditze et al. (1997)		Sc^3+^ > Fe^3+^ > Al^3+^ < Mg^2+^	The organic phase included 20% P204 and 15% TBP in kerosene. The aqueous phase 2.5 g/L Sc, 25 g /L Mg, Al and Fe in 0.5 M HCl	[16]
4	Haslam M et al. (1999)		Sc^3+^ ∼ Zn^2+^ > Ca^2+^ ∼Al^3+^ > Mn^2+^ > Cr^3+^ ∼ Mg^2+^ ∼ Ni^2+^ ∼ Si	The organic phase included 0.2 M P204 and 1% TBP in Escaid 110. The aqueous phase has pH 1.5–3.5 of H_2_SO_4_	[19]
5	Singh RK et al. (2003)		Sc^3+^ > Fe^3+^ > Al^3+^ > Mg^2+^	The organic solution contains 0.1 M P204 in toluene. The aqueous contains 0.5–11 M HClO_4_	[49]
6	Li DQ, et al. (1980)	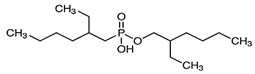 HEHEHP	Sc^3+^ ∼ Th^4+^ > Ce^4+^ > Fe^3+^	The organic solution included HEHEHP in n-heptane. The aqueous is 0.5–1.5 M H_2_SO_4_	[66]
7	Li DQ, et al. (1980)		Sc^3+^ > Ce^4+^ > Th^4+^ > Fe^3+^	The organic solution included HEHEHP in n-heptane. The aqueous is 1.5–5 M H_2_SO_4_	[66]
8	Haslam M et al. (1999)		Sc^3+^ > Zn^2+^ > Al^3+^ > Mn^2+^ ∼ Cr^3+^ ∼ Ca^2+^ ∼ Mg^2+^ > Ni^2+^ ∼ Si	The organic solution is 0.2 M Ionquest 801 and 1% TBP. The aqueous solution has pH 1–5.5 H_2_SO_4_	[19]
9	Singh RK et al. (2003)		Sc^3+^ > Fe^3+^ > Al^3+^ > Mg^2+^	The organic solution is 0.1 M PC-88A in toluene. The aqueous phase is 0.01–1 M HClO_4_	[49]
10	Wang C et al. (1994)	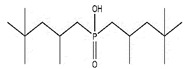 C 272	Sc^3+^ ∼ Th^4+^ > Fe^3+^ > Lu^3+^	The organic solution 4.8 × 10^−2^ M Cyanex 272 in n- hexane. The aqueous phase is H_2_SO_4_ 3–10 M	[53]
11	Haslam M et al. (1999)		Sc^3+^ >> Al^3+^ > Ni^2+^ > Si >Mn^2+^ ∼Mg^2+^∼Ca^2+^> Cr^3+^	The organic solution is 0.1 M Cyanex 272 and 5% TBP. The aqueous phase is H_2_SO_4_ has pH ∼1	[19]
12	Wang C et al. (1995)	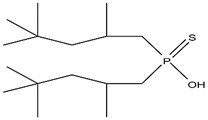 C 302	Zr^4+^ > Sc^3+^ > Th^4+^ > Fe^3+^ > Lu^3+^	The organic solution is 4.8 × 10^−2^ M Cyanex 302 in n-hexane. The aqueous is 2 × 10^−4^ − 6 × 10^−4^ M metals, pH 3–10 M H_2_SO_4_	[76]
13	Wang C et al. (1995)	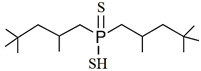 C 301	Zr^4+^ > Sc^3+^ ∼ Fe^3+^ ∼ Th^4+^ > Lu^3+^	The organic solution is 4.8 × 10^−2^ M Cyanex 302 in n-hexane. The aqueous is 2 × 10^−4^ − 6 × 10^−4^ M metals, pH 3–10 M H_2_SO_4_	[76]
14	Wang et al. (2013)	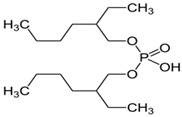 P204 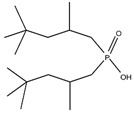 C 272 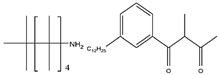 Primene JMT LIX54 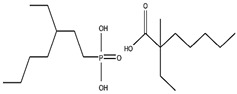 Ionquest 801 V 10	Sc^3+^ < Ti^4+^ < Fe^3+^ < Zr^4^^+^ < Ga	P204 appears to be selective than other reagents while extracting Sc3+ from the leach solution bearing Zr, Fe, Ti & Ga	[57]
15	Peppard DF et al. (1956)	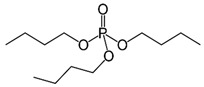 TBP	Sc^3+^ ∼ Zr^4+^ > Th^4+^	The organic solution is 100% TBP, the aqueous is 7–8 M HCl	[77]
16	Zhang et al. (1997)		Sc^3+^ > Zr^4+^	The organic solution is 100% TBP, the aqueous is 4–6 M HClO_4_	[78]
17	Li D et al. (1998)	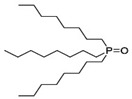 C 923	Zr^4+^ > Sc^3+^ > Ti^4+^ ∼ Lu ^3+^ > Fe^3+^	The organic solution is 5% Cyanex 923 in kerosene, the aqueous is 2.0–7.0 M H_2_SO_4_	[44]
			Sc^3+^ > Th^4+^ > Lu^3+^	The organic solution is 5% Cyanex 923 in kerosene, the aqueous is 1–5 M HCl	
18		NA C 925	Zr^4+^ > Sc^3+^ > Lu ^3+^ > Ti^4+^ > Fe^3+^	2.0–7.0 M H_2_SO_4_	[44]
			Th^4+^ > Sc^3+^ > Lu^3+^	0.5–2.5 M HCl	
19	Onghena et al. (2015)	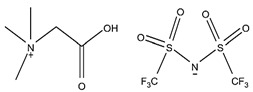 Betainium bis(trifluoromethylsulfonyl)imide, [Hbet][Tf2N],	Sc, Y, La, Ce, Nd, Dy, Fe, Al, Ti, Ca and Na	Sc^3+^ was selectively extracted using [Hbet][Tf2N] from H_2_SO_4_ red mud leachate solution	[79]
20	Baba et al. (2014)	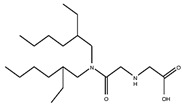 D2EHAG 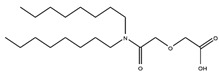 DODGAA 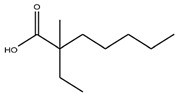 V 10	Sc, Y, La, Nd, Eu and Dy	Sc^3+^ was selectively extracted by D2EHAG due to the chelating effect and the size recognition ability of D2EHAG	[51]
21	Chen et al. (2017)	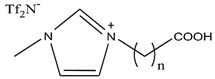 [(CH2)7COOHmim][Tf2N] (n = 3,5,7)	Sc, Y, La	Sc^3+^ extraction was quantitative (99.5%) by [(CH2)7COOHmim][Tf2N] dissolved in [C4mim][Tf2N]	[34]
22	Karve et al. (2008)	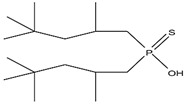 C302	Sc, Y, La, Ce, Pr, Nd, Gd, Dy and Yb	Selective separation of Sc^3+^ was achieved from mixed rare earth leach solution	[35]
23	Fujinaga et al. (2013)	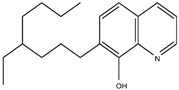 Kelex 100 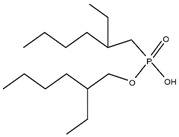 PC88A 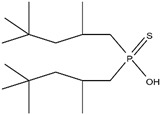 C302 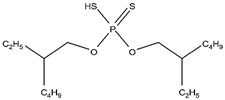 Phoslex DT-8	Sc, Zr, Ti, Y and Al	Extraction of Sc by Cyanex 302 was significantly higher than other extractants; Kelex 100, PC88A	[48]
24	Zaho et al. (2016)	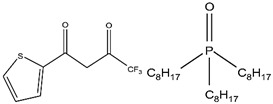 HTTA TOPO	Sc, Al, Fe, Mn, Ni and Zn	Selective extraction of Sc^3+^using HTTA synergism with TOPO	[48]
25	Sun et al. (2007)	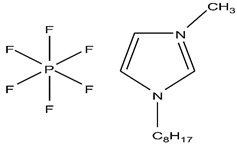 [C8mim][PF6]	Sc, Y, La and Yb	Sc was preferentially separated in presence of Y, La and Yb by the extractant [C8mim][PF6]/Cyanex 925 at A:O = 1:3	[80]
26	Onghena et al. (2017)	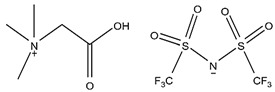 Betainium bis(trifluoromethylsulfonyl)imide, [Hbet][Tf2N]	Sc, Al, Fe	Sulfation-roasting-leaching of Sc from bauxite residue (red mud) followed by selective extraction of Sc with [Hbet][Tf2N] was achieved	[81]
27	Depuydt et al. (2015)	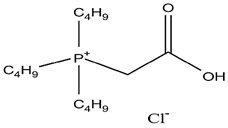 [P444C1COOH] Cl	Sc	The IL-rich phase of the aqueous biphasic system has a very low viscosity, in comparison to the pure IL [P444C1COOH] Cl. This system has excellent extraction properties for Sc	[55]
28	Wu et al. (2007)	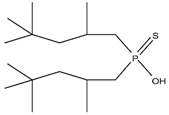 C 302	Sc, Y, La and Gd	Extraction of metals follows the order as Sc > > Y > La > Gd with the extractant Cyanex 302 from their hydrochloride solution.	[82]
29	Kostikova et al. (2005)	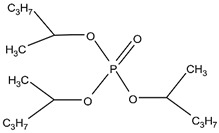 Tri-iso-amyl phosphate (TIAP)	Sc, Zr, Ce, Nd, Sm, Eu, Y, Lu	High-purity Sc can be prepared by multi-step counter current extractive treatment of Sc concentrate with TIAP	[7]
30	S. Das et al. (2018)	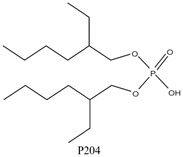 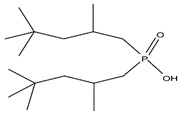 C 272	^Sc^	Extraction of Sc from acidic solutions using organo-phosphoric acid reagents was selective	[69]
31	J. Zhou et al. (2021)	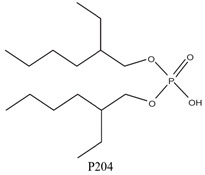 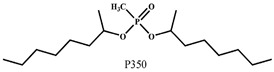 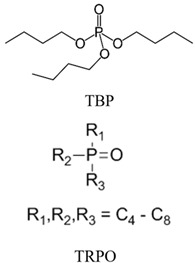	Sc, Ti, Fe, Zr	The mixture of 10% P204 and 5% TBP in kerosene archive extraction efficiency of Sc ~99%	[83]

**Table 2 materials-15-02376-t002:** Some potential approaches for synergistic scandium extraction using chelating extractant mixtures.

Type	Examples	References
Chelating and acidic extractants	Beta-diketone with phosphoric acid, carboxylic acid	[97,98,99]
Chelating and solvating extractants	Beta-diketone with phosphate, phosphine oxide, phosphine sulphide, and sulphoxide	[92,93,94]
Chelating and basic extractants	Beta-diketone with aliquat chlorides	[95,96]

**Table 3 materials-15-02376-t003:** Comparison of solvent extraction by OPCs for Sc recovery from red mud.

Extraction Step		Stripping Step		Year/References
Extractant	E%	Stripping Agent	S%	
15% D2EHPA 5% TBP	99.00	2 M NaOH	95.40	(2017) [157]
8% D2EHPA 2% TBP	99.70	2 M NaOH + 1 M NaCl	85.00	(2018) [158]
16% D2EHPA 4% TBP	99.00	2 M NaOH	96.00	(2019) [159]
100% TBP	99.00	Pure water	-	(2019) [160]
60% Cyanex 272 40% Cyanex 923	98.00	10% H_2_C_2_O_4_	98.80	(2020) [59]
15% D2EHPA 15% N1923	99.00	5 M HNO_3_	89.30	(2020) [42]
10% D2EHPA 5% TBP	99.00	5 M NaOH	99.61	(2021) [161]
10% D2EHPA 5% TBP	99.00	3 M NaOH	99.00	(2021) [162]

## Data Availability

Data is contained within the article research.

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
