# Peer review of "Scandium Recovery Methods from Mining, Metallurgical Extractive Industries, and Industrial Wastes"

_materials, 2022, doi:10.3390/ma15072376_

Round 1

Reviewer 1 Report

Scandium is a very important rare metal. In recent years, extracting scandium from various wastes to realize resource cycle has attracted widespread attention. Many researchers have developed very meaningful methods and approaches. There's a lot of innovative research going on. It is necessary to review the cycle extraction of scandium. This paper focuses on the review of scandium extraction solution and liquid scandium extraction methods, as well as the use of various extraction agents, these content is good; At the same time, the process of extracting scandium from different resources and wastes, such as rare earth ore, uranium ore, titanium and zirconium ore, nickel ore and red mud, is listed. All of these have certain reference significance, but there are areas that need to be optimized and improved: 1. The title of the paper is mainly about the review of scandium extraction from red mud, but in fact, the content of scandium extraction from red mud only accounts for a small part at the end of the paper, and most of the paper focuses on the separation and extraction of scandium in solution and from other minerals, not red mud. There are some misericordities, and the logic and organization of the text is disorganized. 2. Although there are 147 references in this paper, the author does not cite several good core articles about scandium extraction from red mud, such as: [1] Liu, Zhaobo, Yanbing Zong*, Li, Hongxu*, Zihan Zhao; Characterization of scandium and gallium in red mud with Time of Flight-Secondary Ion Mass Spectrometry (ToF-SIMS) and Electron Probe Micro-Analysis (EPMA) . Minerals Engineering ,Volume 119, April 2018, Pages 263-273 [2] Zhaobo Liu, Hongxu Li*, Qiankun Jing, Mingming Zhang. Recovery of Scandium from Leachate of Sulfation-Roasted Bayer Red Mud by Liquid–Liquid Extraction.JOM (The Journal of The Minerals, Metals & Materials Society), November 2017, Volume 69, Issue 11, pp 2373–2378

Author Response

Please find point-by-point response in the attached document! Thank you for your efforts to improve our manuscript!

Reviewer 2 Report

Scandium Recovery from Mining, Metallurgical Extractive Industries and Industrial Wastes: A review on bauxite waste valorization

 Congratulations to the authors of the paper. A very well-written, substantive article that can be an encyclopaedic guide / review on the recycling of Sc from various industrial waste. The study provides a wide, comprehensive knowledge (as for a scientific journal) on the recycling of a very important critical / rare metal through valorization of post-production and post-consumer waste. Paper provides an up-to-date, comprehensive literature review

Remarks

  1. Title of paper.

 In Introduction:

 Furthermore, the metallurgical procedures for scandium recovery from a variety of sources, particularly from the bauxite wastes (red mud) are the primary objectives.

So in my opinion the title should be

  Review of Scandium Recovery Methods from Mining, Metallurgical Extractive Industries and Industrial Wastes

and add  RM as one of keywords

 Keywords: Scandium recovery, Hydrometallurgy, Solvent extraction, Organophosphorus extractants, Red Mud

  1.  

Page 3

 High-intensity metal halide lamps, analytical standards, electronic components, oil well tracer, fuel cells [9]. 

Something is missing in this task, isn't it an ending to the next sentence?

The production of high-purity materials has been highly attracted by recent scandium uses, for laser research [1,10].

3.

In aluminum, cobalt, iron, molybdenum, nickel, phosphate, tantalum, tin, titanium, tungsten, uranium, zinc and zirconium it also coexists widely in low content [1,4,1].

 Is it about metals or ores of these metals containing scandium?

4.

 Page 4

 form), (2) SX to increase the amount of desirable minerals, and (3) purification of the

Rather chemical compunds than minerals arise in such extraction methods.

  1.  

 page 5

 Figure 2. Chemical structures of the acidic phosphorus extractants A. phosphoric acid, B.

 lack of a colon

 Figure 2. Chemical structures of the acidic phosphorus extractants:  A. phosphoric acid, B.

Author Response

(The authors gave the same response as above.)

Reviewer 3 Report

This article is a mixed description of extractants and scandium-containing raw materials and has limited relevance to the Materials journal. The review does not correspond to the journal's theme, and its content is more appropriate for the journal “Hydrometallurgy” or “Minerals”. But the main thing in this form the review cannot be published since the check for anti-plagiarism showed that the originality of the article is only about 60.6%, which is unacceptably low even for a review.

Low originality is because the work duplicated the information presented in papers 1 and 2.

Half of the listed works (about 77) were published before 2011, and therefore, the work has low originality.

In addition, the work is rife with errors and inaccuracies. Below are examples:

  1. Some sources do not fit the topic, for example, cited works where there is no information about Sc or the reference does not match. For example, in works 16.17, nothing is said about the supply of scandium obtained during the Cold War. And the phrase itself is incorrect since scandium was not obtained in Russia, but in the USSR, more precisely in Ukraine.
  2. Table 1 is taken from 2 and 9 sources with errors in the references (for example, they refer to the work about rKa, although this is not the case). references appear not to correspond to the order of numbering (86, 87 in Table 1 appeared before the appearance of 70 references in the text).
  3. [Peppard]'s source is unnumbered in the reference list, so the numbering is off.
  4. Errors with references in the text. e.g., Kim [119,120] indicated that about 100%. The author owns one of the works in the reference list.
  5. According to Wand and Cheng [2], scandium in... error in the authors of the paper. Wang and Cheng refer to the work of other authors.
  6. The presented numbers of extraction ability are given with errors. In two rows there is a repetition of elements:

 Fe3+> Lu3+> Lu3+> Yb3+> Er3+> Y3     

Th > Ti > Zr > Hf > Sc > U > Sc.

Author Response

(The authors gave the same response as above.)

Reviewer 4 Report

In the manuscript with the title “Scandium Recovery from Mining, Metallurgical Extractive Industries Wastes: A review on bauxite waste valorization”, the authors are reporting the metallurgical extractive procedures for scandium recovery.

There are however a few aspects that have to be clarified before publication and revision to some extend is required.

  1. Introduction:

In aqueous solution the stable oxidation status of scandium is scandium [5].

Please refer which is the stable oxidation state of scandium.

2.1.2 Carboxylic Acids

In Table 1 please use one font type and size. Some of the structures are “stressed” and blurred.

 There are some missing details in the references (bold, font type, etc.). Furthermore, you will need to refer to more recent bibliography.

Overall, I think that this article contains enough robust data.

Following minor revision, I believe that this paper can be considered for publication.

Author Response

(The authors gave the same response as above.)

Round 2

Reviewer 1 Report

It has been modified according to opinions.

Author Response

The authors would like to thank the editor and reviewers for their detailed and constructive comments and suggestions for the submitted manuscript. The authors strongly believe that the comments have identified essential areas that required improvement in the submitted paper. After completion of the suggested edits, the revised manuscript has benefitted from an improvement in the overall presentation and clarity.

Reviewer 3 Report

Review 2

Despite corrections and additions of some sources, the originality of the article has not changed significantly. As before, several dozen links from 1 (Wang, W.; Pranolo, Y.; Cheng, C. Y., Metallurgical processes for scandium recovery from various resources: A review. Hydrometallurgy 2011, 108 (1), 100-108) and 2 (Wang, W.; Cheng, C. Y., Separation and purification of scandium by solvent extraction and related technologies: a review. Journal of Chemical Technology & Biotechnology 2011, 86 (10), 1237-1246) sources are duplicated in the list of publications. And the conclusions of the article are 75% the same as the conclusions of the review in Hydrometallurgy. It is also necessary to justify the expediency of duplicating the schemes from the article of the journal Hydrometallurgy. (W. Wang, Y. Pranolo, C.Y. Cheng. Metallurgical processes for scandium recovery from various resources: A review. Hydrometallurgy 108 (2011) 100–108). 

The relevance of the review, in which much attention is paid to extraction processes, is reduced since in 2021 a review article “A review on solvent extraction of scandium” was published in the Journal of “Rare Earths” (https://doi.org/10.1016/j.jre.2021.12.009) to which there is no reference in the work. Also, not mentioned in this work is the 2021 paper "A systematic study on extraction and separation of scandium using phosphinic acid by both solvent extraction and hollow fiber (DOI: 10.1080/25726641.2021.1908079).

It is recommended not to repeat most of the references from works 1 and 2 and it is better to justify the need to publish your work.

Author Response

The authors would like to thank to the editors and reviewer for their detailed and constructive comments and suggestions for the submitted manuscript. The authors strongly believe that the comments have identified essential areas that required improvement in the submitted manuscript. After completion of the suggested edits, the revised manuscript has benefitted from an improvement in the overall presentation and clarity.

In the attached document, the reviewer can find a point-by-point description of how each comment was addressed in the revised version of the manuscript. In this reviewer report, the original reviewer comments are written in black; the corresponding responses are given in blue. The corresponding changes in the revised version of the manuscript are mentioned in green color background.

The novelty of our paper

Despite that are a lot of previous published papers related to the scandium (according to our literature review), no any review comparatively presented in deep details the processes involved in both mining and recovery of scandium from industrial wastes/by-products. The revised manuscript covers the most recently used solvent extraction techniques for scandium recovery, using different extractants provided by various producers. We have presented the most important previous published papers related to the scandium recovery from various sources (ores and waste) since 1969 up to 2022. We hope that this review could be useful for all which are interested in scandium recovery at industrial scale from all existing scandium sources, providing a solid information background for such kind of applications.

The main changes in the revised manuscript based on reviewer comments:

  1. All references were rearranged and updated based on editors and reviewer comments.
  2. All figures were modified, and others two new figures were added (Figures 21 and 18).
  3. New updated references (2019-2022) were added, being in the manuscript text highlighted with green color.
  4. A new subchapter was added with titled “Limitations, challenges and future research directions for Scandium Recovery” which include our original contribution based on both literature review and our experiences accumulated during a few running projects related to this topic.
  5. Both the Abstract and Conclusions were updated based on our scientific discussions related to the scandium recovery from the red mud, the main important industrial waste in huge quantities, which have an important content of scandium and led to the environmental issues due to their disposal.
  6. A new table (Table 3) was included which summaries and describes the last articles related to the scandium recovery from the red mud by solvent extraction using OPCs (organophosphorus compounds).
  7. Please note the updated References: 10, 12, 32 , 42, 45, 59, 116, 145, 146, 147, 156, 160 , 161, 162, 163, 164, 165, 166, 169.

Round 3

Reviewer 3 Report

The number of references to literature older than 10 years could be reduced and links to reviews could be provided.